# The importance of cognitive diversity for sustaining the commons

Jacopo A. Baggio[1,2], Jacob Freeman[3,4], Thomas R. Coyle[5], Tam The Nguyen[6], Dale Hancock[5], Karrie E. Elpers[5], Samantha Nabity[3], H.J.Francois Dengah II[3] & David Pillow[5]

Cognitive abilities underpin the capacity of individuals to build models of their environment and make decisions about how to govern resources. Here, we test the functional intelligences proposition that functionally diverse cognitive abilities within a group are critical to govern common pool resources. We assess the effect of two cognitive abilities, social and general intelligence, on group performance on a resource harvesting and management game involving either a negative or a positive disturbance to the resource base. Our results indicate that under improving conditions (positive disturbance) groups with higher general intelligence perform better. However, when conditions deteriorate (negative disturbance) groups with high competency in both general and social intelligence are less likely to deplete resources and harvest more. Thus, we propose that a functional diversity of cognitive abilities improves how effectively social groups govern common pool resources, especially when conditions deteriorate and groups need to re-evaluate and change their behaviors.

---

[1] Department of Political Science, University of Central Florida, Orlando, FL 32816, USA. [2] Sustainable Coastal Systems Cluster, National Center of Integrated Coastal Research, University of Central Florida, Orlando, FL 32816, USA. [3] Department of Sociology, Social Work, and Anthropology, Utah State University, Logan, UT 84322, USA. [4] Ecology Center, Utah State University, Logan, UT 84322, USA. [5] Department of Psychology, University of Texas at San Antonio, San Antonio, TX, USA 78249. [6] Department of Computer Science, Utah State University, Logan, UT 84322, USA. Correspondence and requests for materials should be addressed to J.A.B. (email: jacopo.baggio@ucf.edu)

In the biological and the social sciences there are few topics as important as the consequences of diversity for the functioning and transformation of ecosystems and social systems[1–4]. One of the most important lines of research in ecology, for instance, is the effect of diverse functional traits on the stability and efficiency of ecosystems[1,5]. Functional traits include physiological, morphological, and phenological traits that affect individual fitness[6,7] and provide mechanistic insights into how species may respond to disturbance. A system with higher functional diversity and redundancy of functions allows ecosystems to withstand disturbance and maintain a consistent level of productivity[1,5,8–11]. For example, fisheries with a greater diversity of functional groups produce a higher level of fish biomass more consistently than less diverse fisheries[12]. While the consequences of a functional diversity of traits are well understood for ecosystems, the effects of functional traits within cognitive and social contexts on the governance of natural resources is less well developed.

Akin to functional traits in ecology, cognitive functional traits, such as general ($g$) and social intelligence, specifically, theory of mind (ToM), are domain general mental abilities that allow individuals to process information and adapt in social-ecological settings. $g$ reflects the variance common to mental tests (e.g. IQ tests) and measures the ability of individuals to engage in complex reasoning and abstract thought[13]. ToM is the ability to model and reason about the intentions of others[14–16]. Given these definitions and the different tasks that $g$ and ToM help individuals accomplish, we postulate that a functional diversity of intelligences improves the ability of groups to govern resources and that intelligence functional diversity is maximized when groups have high competency in both $g$ and ToM. In particular, in this paper, we investigate the effects of cognitive functional diversity on the ability of social groups to govern a common pool resource system.

A common pool resource system is a system in which resources are non-excludable (all individuals have access) and the harvest decisions of each individual affect the availability of resources for the entire group (i.e., the resource is rivalrous). In such systems, governance entails developing rules and norms that allow individuals to harvest the resource now and, at the same time, create incentives for sharing and preserving the resource for future generations[17,18]. In order to assess the relationship between a group's ability to sustain common pool resources and cognitive abilities, we conduct behavioral experiments in a spatially explicit common pool resource system to test the functional intelligences proposition (FIP)[19].

The basic premise of the FIP is that $g$ and ToM serve different functions. This statement is supported, first, by the fact that high $g$ individuals with autism show a deficit in ToM[14], and when such high ability individuals attempt to model others' mental states, brain regions associated with ToM remain inactive[14]. Second, while aspects of $g$ (e.g., cause and effect detection and transitive inference) are pervasive among vertebrates[20], evidence of ToM remains rare[20]. The best evidence of ToM among nonhuman vertebrates comes from social animals with complex communication systems, in particular, blue jays, ravens, and chimpanzees[21]. These patterns suggest a widespread convergent selective pressure for the ability to detect cause and effect and reason about such relationships (aspects of $g$) among vertebrates, but this selective pressure does not, apparently, lead to widespread ToM abilities. In other words, the two abilities are not co-evolving in nature. Finally, direct measures of social-cognitive ToM weakly correlate with measures of $g$ among human populations[22].

Given that $g$ and ToM serve distinct functions, both abilities should affect the performance of social groups that attempt to harvest common pool resources. This is because harvesting common pool resources requires both a cognitive mapping of resource dynamics and effective models of others' intentions. In particular, we propose that the highest functional diversity of intelligence occurs when groups contain individuals with high $g$ and high ToM. In this scenario, groups are made up of individuals good at mapping their biophysical environment and individuals good at mapping and communicating intentions within their social environment. In short, $g$ and ToM should interact, and, as both abilities increase, groups should more effectively solve the collective action dilemmas that arise in a common pool resource system. However, if either capacity declines, collective action becomes more problematic and thus the sustainable management of resources more difficult.

Collective action dilemmas continually arise in a common pool resource system because of the incentives that an individual has to maximize her gains while dealing with uncertainties related to resource abundance and the behavior of other individuals in the system. Such uncertainty, along with the willingness to maximize one's own gain, creates an incentive to over exploit resources. Following Hardin's seminal work[23], only two strategies were once thought capable of solving such dilemmas and conserving common pool resources: (1) strong, top–down state control, and (2) privatization[23]. However, the depletion of common pool resources does not inevitably occur absent state control or private property rights[18]. In fact, researchers have documented multiple cases in which groups sustainably manage common pool resources from the bottom-up[17,18,24–27]. These groups often display five common characteristics: They (1) adapt rules of harvesting and resource appropriation to local resource dynamics; (2) establish a proportionality between the provision and appropriation of resources; (3) monitor the resource itself; (4) sanction those who do not comply with the community rules (or the rules of the commons); and (5) clearly define who has access to harvesting resources[17].

In the context of the five characteristics above, $g$ is critical to understand the local resource dynamics[13,28]. Higher $g$ should allow groups to analyze and assess resource changes, hence the higher the total group $g$ (or the average), the more likely a group contains many individuals who model resource dynamics correctly and notice changes in local resource conditions[29]. However, higher $g$ also implies an increase in individuals who rationally calculate the costs and benefits of using resources and more readily compute strategies that will yield the highest net benefit to themselves[30]. Thus, the effects of $g$ on the harvest of common pool resources should remain context dependent. For example, harvesting resources in systems where boundaries, monitoring of resources and rule matching are irrelevant, high $g$ individuals should maximize their own short-term rewards[31]. However, when harvesting resources in common pool resource systems, higher $g$ individuals assess uncertainty in both the resource (abundance and dynamics) as well as in others' behavior. Especially when uncertainty related to others' behavior is high, high $g$ individuals are more likely to defect in order to maximize their current benefit, potentially leading to the overharvest of resources[32].

ToM is critical for individuals to model and monitor others' mental states and social positions[14,15,33]. Higher ToM should increase an individual's ability to anticipate and monitor others' behaviors, abide by inclusive rule making and diffuse more efficiently, either through their actions or words, conflicts that may arise in common pool resource systems[33–36]. A higher ability to model others' mental states is associated with more efficient social interactions and more pro-social behavior as defined by Frey[37], allowing groups with higher ToM to build and maintain fair and legitimate rules that take proportionality and local circumstances into account. Intelligence research in psychology also indicates that increases in ToM improves the ability of groups to achieve a

mutually beneficial goal in a static environment. For example, Woolley and colleagues[36] find that the $g$ of individuals does not predict a general group level intelligence factor but ToM does[35,36]. This means that groups with higher ToM perform better on a battery of tasks than groups with lower ToM scores. In contrast with $g$, where the sum of the ability of individuals forming a group drives the overall understanding of a system, any one individual with low ToM may jeopardize the whole group's ability to devise effective rules to manage the system[29,38], reducing a group's ability to manage resources in the face of change and, in some cases, leading to overharvest.

In the case of environmental change that affects resources, a functional diversity of cognitive abilities should be critical for adapting to negative changes (HL treatment –discussed below), while, perhaps, not as important when environmental change improves a resource (LH treatment–discussed below). The difference in the importance of cognitive abilities may stem from the difference between resource and group dynamics when conditions improve vs. degrade. When conditions improve, there is no need for re-negotiation and one can keep behaving as she did in the past without adverse consequences. On the other hand, when conditions deteriorate, harvest behavior needs to change in accordance with the new condition of local scarcity. In this context, negotiations about resource appropriation need to happen in order for the group to continue to manage resources sustainably. For example, in repeated and environmentally stable situations, all groups may eventually find optimal solutions (i.e., learning by doing). However, when the ecological system changes, effective mental models of the underlying resource dynamics, as well as other individuals' behavior are critical. More effective information processing improves learning and adaptation. Groups composed of individuals with higher $g$ should better adapt to changes in their resource base than groups with lower $g$, as groups with higher $g$ more readily detect changes in a system and devise rules to match such changes. However, changes in the biophysical resource system often require the re-negotiation of social rules and the communication of new knowledge. Hence, higher ToM should increase a group's ability to work towards a common goal.

In sum, the combination of high $g$ and high ToM should lead to groups who are better at solving collective action dilemmas and, thus, managing a common pool resource system[19]. Figure 1 summarizes the predicted interaction between $g$ and ToM. In the lower right quadrant, groups composed of many individuals with high $g$ should effectively monitor a resource and develop rules that match the dynamics of the resource. However, such groups also have low ToM and should experience more conflict, especially when a system changes. The costs associated with conflict should nullify the gains from higher $g$. Another way to think about this prediction is that ToM affects how efficiently individuals cooperate in groups and may lead to the emergence of a general group intelligence factor[39]. When ToM is low, a group has a lot of individual cognitive capital relevant to understanding the resource, but conflict and difficulty communicating blunt the emergence of a group level intelligence that harnesses the individual level capital to maintain good governance over time. Similarly, in the upper left quadrant, groups with high ToM but low $g$ should lack individuals with effective mental models of how a resource system works, and this may nullify the benefits of more amicable groups. In the lower left quadrant, low $g$ and ToM groups should find it difficult to understand the resource system and effectively work together, which should lead to poor governance. Finally, in the upper right quadrant, high $g$ and ToM should improve governance, especially in a system susceptible to negative changes in a resource base over space and time. Groups that contain individuals with high $g$ and high ToM understand

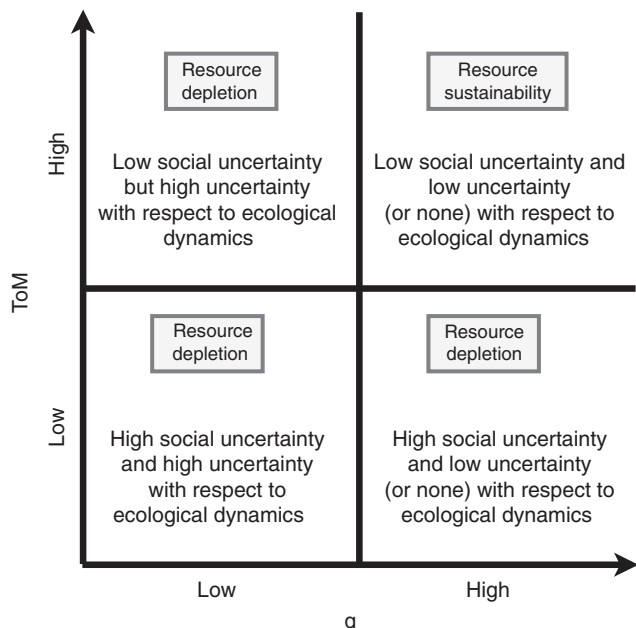

**Fig. 1** Predicted interaction effects of $g$ and ToM on the ability of groups to harvest resources and adapt to changes in resource growth

the system well and more efficiently work together to govern a resource.

Our results indicate that groups with high $g$ and ToM better adapt to deteriorating environmental conditions. Such groups are less likely to deplete resources and harvest more resources, as these groups have a better understanding of how the system works and are also able to negotiate and communicate effectively. Conversely, our results also indicate that when conditions improve, groups with high competency in $g$ more effectively reap the benefit of the positive change. In fact, high $g$, along with reciprocity, is sufficient for groups to perform well when resource conditions improve, as conflict situations are less likely to arise when resources are plentiful. In this situation the discriminating variable between group performances is how well each group understands the resource system.

## Results

**Assessing effects of $g$ and ToM**. To evaluate the effects of a functional diversity of intelligence capacities, we ran behavioral experiments by modifying the experimental environment developed by Janssen and colleagues[40]. The experimental environment consists of a spatially dispersed resource (tokens) that grows according to a density-dependent function (see Supplementary Method 5). In this environment, groups of four harvest tokens for six rounds. Two experimental treatments manipulate the growth function to simulate a perturbation to the resource base of the common pool resource (see Methods and Supplementary Method 5). In treatment one, groups harvest resources at a high growth rate for three rounds and then experience a sudden decline in the growth rate of the resource and harvest resources with a lower growth rate for another three rounds (HL treatment). In treatment two, the exact opposite sequence occurs (LH treatment). In both treatments, groups harvest tokens with a specific growth rate for three rounds (rounds 1–3), and then with a changed growth rate for another three rounds (rounds 4–6). The change always occurred between rounds 3 and 4.

To isolate the effects of $g$ and ToM, we use three response variables. (1) The time a group stares at a collapsed resource in a given round (Time). The greater the proportion of time within a

round that a group stares at an empty screen, due to complete resource depletion, the less effectively they governed the common pool resource. (2) The difference in the percentage of potential tokens harvested, on average, before and after an ecological change ($\Delta T$, see also Supplementary Method 1 for details). The lower (or negative) this difference, the less intense the harvest pressure on the resource after the change in resource growth. (3) The percentage of potential tokens harvested per round (avg$T$, see also Supplementary Method 1 for details). This metric estimates how fully each group uses the resource round-to-round. The three metrics used are related, and assess slightly different group performance metrics: Time relates to resource depletion and collapse; $\Delta T$ relates to harvest pressure pre vs.post environmental change, and avg$T$ relates to the ability of groups to harvest at the optimal level.

Based on previous work, we first assess the effect of $g$ and ToM on Time and $\Delta T$ controlling for factors known to affect the ability of groups to sustainably manage common pool resources: (1) the volume of communication during the experiment, (2) a self-reported measure of trust, (3) ethnic diversity, (4) religious diversity, (5) the proportion of males in each group, and (6) reciprocity[40–46]. Communication and trust often have positive effects on sustaining a common pool resource[40–43]. Cross-cultural studies suggest that within-group ethnic and religious homogeneity[45,46] have positive effects on the governance of resources. Finally, previous work also indicates that gender composition may also affect the ability of groups to manage common pool resources[43], and that individuals are more likely to cooperate in the future rounds of repeated games if they cooperated in previous rounds[42,47]. Controlling for these factors, we find that groups with high $g$ and high ToM manage a common pool resource better than groups lacking in one or both of these distinct cognitive abilities.

Secondly, we assess a three-way interaction effect of $g$, ToM and ecological change on avg$T$ in order to assess how cognitive abilities affect a group's ability to follow the best cooperative protocol for maximizing the number of tokens collected. When environmental conditions deteriorate (HL treatment), we find, once again, that groups with both high $g$ and high ToM harvest closer to the optimal level than groups lacking in one or both of these distinct cognitive abilities.

**Interaction effect of $g$ and ToM on governing the commons.** To evaluate the effects of $g$ and ToM on the ability of groups to manage common pool resources, we used a general linear model by re-scaling the dependent variable Time in the [0,1] interval. Where both 0 and 1 have a theoretical positive probability of an actual outcome: 0 = a group is able to harvest tokens until the end of a round, and 1 = a group collapses the resource immediately. We ran six models per treatment. For example, model HLt1 regresses Time on $g$, ToM, $g$*ToM and the results of the previous round. Model HLt2 includes all of the independent variables of HLt1, plus communication volume and so on until model HLt6, which includes all of the additional factors (see also Supplementary Method 2 and Supplementary Table 2).

In both the high-to-low (HL) and low-to-high treatments (LH), the performance of a group in the previous round always has a significant effect on the current round (see Supplementary Method 2 and Supplementary Table 2 for more details). For example, if groups sustain their resource until near the end of round two, they are more likely to do so in round three as well. In the LH treatment, performance in the previous round is the only significant factor affecting Time. This may occur because beginning the game with very low resource growth sends an unambiguous signal to individuals that the resource is on the edge

of collapse. This lack of ambiguity about the consequences of harvesting too fast could, potentially, impact the importance of cognitive factors in managing the resource (i.e., the problem is simple). However, in the HL treatment, in addition to the previous round variable, $g$ and ToM have a significant interaction effect.

The interaction effect of high $g$ and high ToM is illustrated by Fig. 2 (see also Supplementary Method 2, Supplementary Table 2 and Supplementary Fig. 7). Figure 2 portrays a series of marginal effect plots (one for each regression model run in each treatment). The dark blue shading is the parameter space in which groups of corresponding $g$ and ToM values have a higher probability of harvesting resources up to the end of a round, and, as the shading becomes lighter and redder, groups have a higher probability of spending more time staring at an empty screen as resources deplete. Figure 2 illustrates that groups with both high $g$ and high ToM are less likely to collapse their resource, especially in the HL treatment, before a round ends. However, Fig. 2 indicates that increases in $g$ or ToM alone increase the probability of over-harvesting (see also Supplementary Table 2). For example, in the plot labeled HLt1, the dark blue is in the upper right hand corner of the graph where $g$ and ToM are both high. That is, groups with high $g$ and high ToM use resources more sustainably. However, low ToM and high $g$ (bottom right corner of a given plot) or high ToM, low $g$ (top left corner of a given plot), lead to a higher probability of collapsing the resource well before a round completes. This pattern is displayed in both treatments; however, the effect is weaker in the LH treatment and only significant in the HL treatment (see Supplementary Method 2, Supplementary Table 2). These results are further confirmed by analyzing the interaction between cognitive abilities and the ecological change (see Supplementary Method 3, Supplementary Table 2 and Supplementary Fig. 8). The interaction term is, once again, significant and negative, while $g$ and ToM alone have a positive effect on Time. In other words, groups with both high $g$ and high ToM, are more likely to avoid resource depletion. Conversely, high $g$ and low ToM or vice-versa is associated with a faster collapse of the resource (Fig. 2, Supplementary Method 2, Supplementary Table 2, Supplementary Method 3, Supplementary Table 4 and Supplementary Fig. 8).

To follow up on the results presented above, we ran an OLS regression employing intelligence ($g$, ToM, and $g$*ToM) to predict changes in the percentage of the maximum potential tokens harvested before and after an ecological change: $\Delta T$ (see Fig. 3, and Supplementary Table 3). This allows us to track, on average, whether cognitive abilities influence groups' responses to ecological changes by increasing (i.e. harvest more tokens after a change, $\Delta T > 0$), reducing (i.e. harvest fewer tokens after a change, $\Delta T < 0$) or maintaining constant pressure ($\Delta T = 0$). Analogous to above, we ran six models per treatment, one that includes just $g$, ToM & $g$*ToM and five that include control variables (see Supplementary Method 2 and Supplementary Table 3 for more details). We find that across all regression models $g$ and ToM have positive, statistically significant independent effects on the response variable $\Delta T$ (Supplementary Table 3). This means that where either $g$ or ToM is higher alone, groups harvest more of the potential tokens from the resource system after an ecological change and, thus, push the system closer to a potential collapse. However, $g$ and ToM also have a negative, statistically significant interaction effect on $\Delta T$. This indicates that when $g$ and ToM are higher together, resources are harvested more sustainably after an ecological change. In short, after a perturbation changes the growth rate of the resource, groups with high $g$ or high ToM alone push the resource closer to its ecological limit (increase harvest pressure). The interaction of $g$ and ToM compensates for these

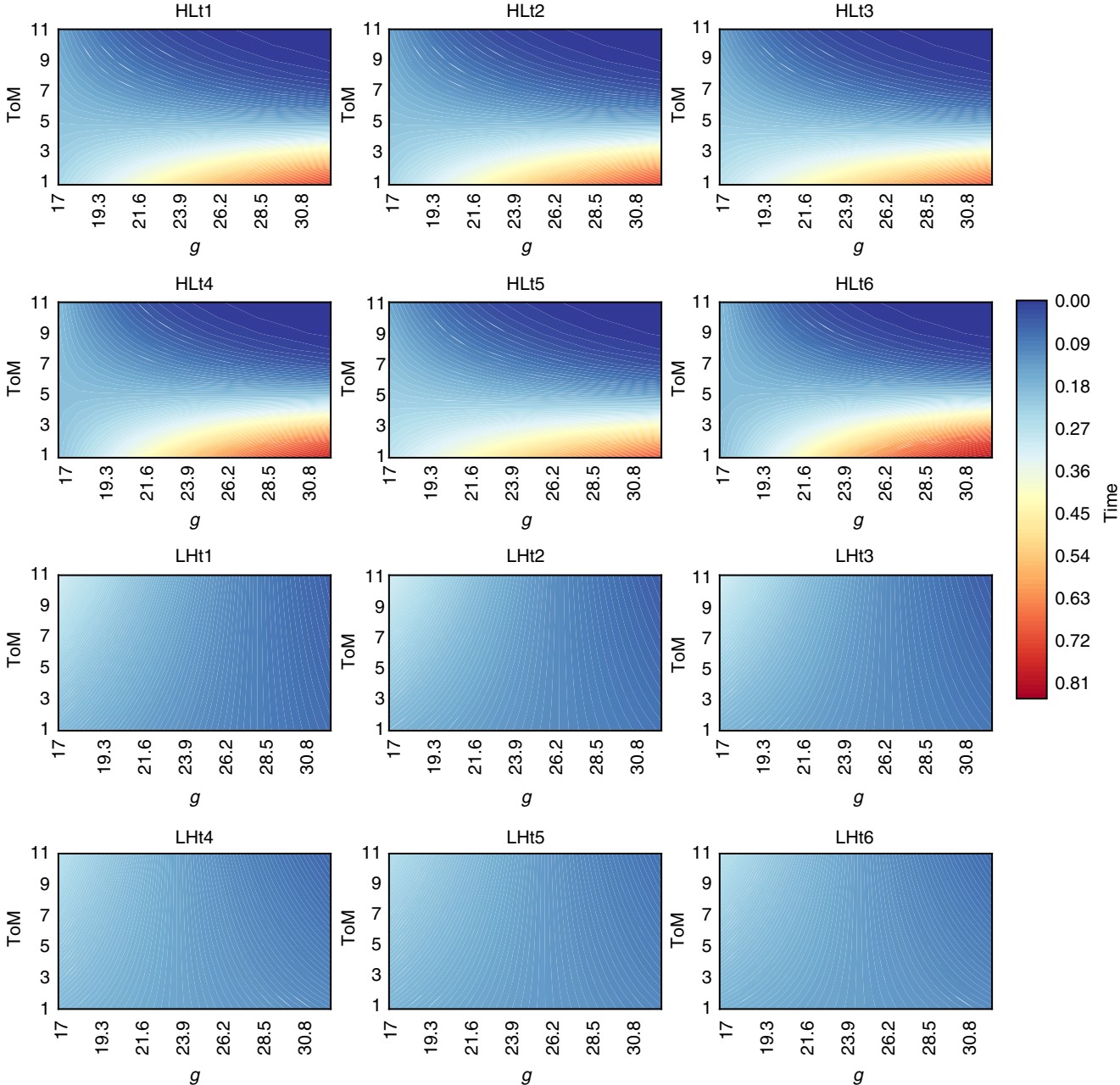

**Fig. 2** Effects of $g$ and ToM on Time. Blue color = group harvested for more time (i.e. did not or collapsed the resource later). LH low-to-high resource growth treatment, HL high-to-low resource growth treatment. The different figure panels represent the marginal effects of $g$ and ToM for the different models portrayed in Supplementary Table 2

individual effects of $g$ and ToM, indicating that groups with high values of both cognitive abilities more readily adapt to a perturbation to their resource system and maintain a more sustainable harvest.

These results are further supported by analyzing the ability of groups to harvest resources near the optimal level for a simulated group (avg$T$). In this regression analysis, we include a three-way interaction between $g$, ToM, and ecological change (see Supplementary Method 3 and Supplementary Table 4). avg$T$, the response variable, is defined as the percentage of the maximum potential tokens harvested per round by a group (avg$T$), and change is a dummy variable that assumes a value of 0 in rounds 1–3 (pre growth-rate change), and a value of 1 in rounds 4–6 (post growth-rate change). Figure 4 illustrates, as above, the marginal effects of $g$ and ToM, and the importance of

both high $g$ and high ToM for sustainably managing the resource in the HL treatment. Panels HLr4-b and HLr4-a indicate the marginal effects of $g$ and ToM before (-b) and after (-a) the ecological change in the HL treatment. These panels illustrate that groups with higher competency in both cognitive abilities harvest closer to the optimal level (indicated by the dark blue color on the top right corner of panels HLr4-b and HLr4-a). When both $g$ and ToM are high, groups harvest a greater percentage of potential tokens because they cooperate better and do not collapse the resource base as quickly as groups with only high $g$ or high ToM. In contrast, prior to the resource change in the LH treatment, groups with high $g$ do better than groups with high $g$ and high ToM or just as well. In other words, when conditions improve, groups with high $g$ are better able to take advantage of the improved conditions and

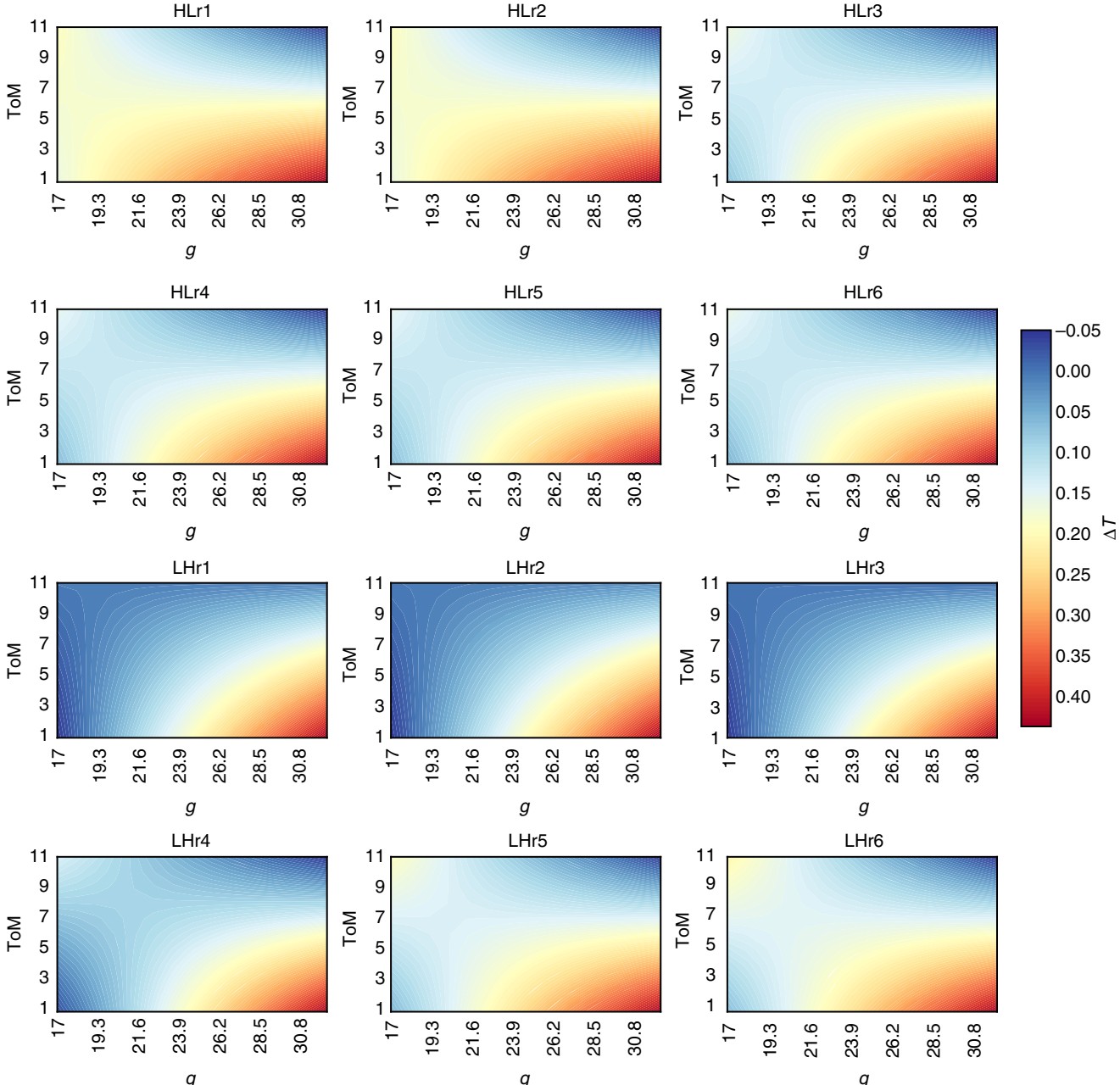

**Fig. 3** Marginal effects of $g$ and ToM on $\Delta T$. Blue color less change between the first three and the second three rounds, LH low-to-high resource growth treatment, HL high-to-low resource growth treatment. The different figure panels represent the marginal effects of $g$ and ToM for the different models portrayed in Supplementary Table 3

hence harvest closer to the optimal level (the dark blue is in the lower right-hand corner of all effect plots labeled LH in Fig. 4, and Supplementary Fig. 9).

In sum, when conditions deteriorate (HL treatment) Figs. 2–4 illustrate that groups with high $g$ but low ToM push a system closer to its ecological limits, are more likely to deplete resources faster, and harvest a lower percentage of potential tokens. Groups high in both $g$ and ToM put less harvest pressure on the resource base, are more likely to avoid resources depletion altogether, and harvest a greater percentage of tokens. When the ecological change is positive (LH treatment), groups with higher $g$ tend to harvest a greater percentage of potential tokens. This most likely occurs because these groups realize the novel opportunity presented by the decreased likelihood of resource depletion in that environment

## Discussion

This work contributes to identifying how a functional diversity of cognitive abilities affects the ability of groups to manage resources sustainably. It is well understood that functional diversity increases the stability of ecosystems faced with disturbances[1]. The effects of diversity in social systems are more contextual[48–52]. For example, some argue that less ethnic and religious diversity promotes the governance of resources because homogeneity reduces uncertainty within the social domain and, at the same time, may increase trust[53,54]. On the other hand, such diversity also generates different learning heuristics and perspectives, which may increase the ability of social groups to innovate and adapt to social change[2,55]. Previous studies also indicate that personality and behavioral diversity have long-term positive effects on the performance of teams while the benefits due to

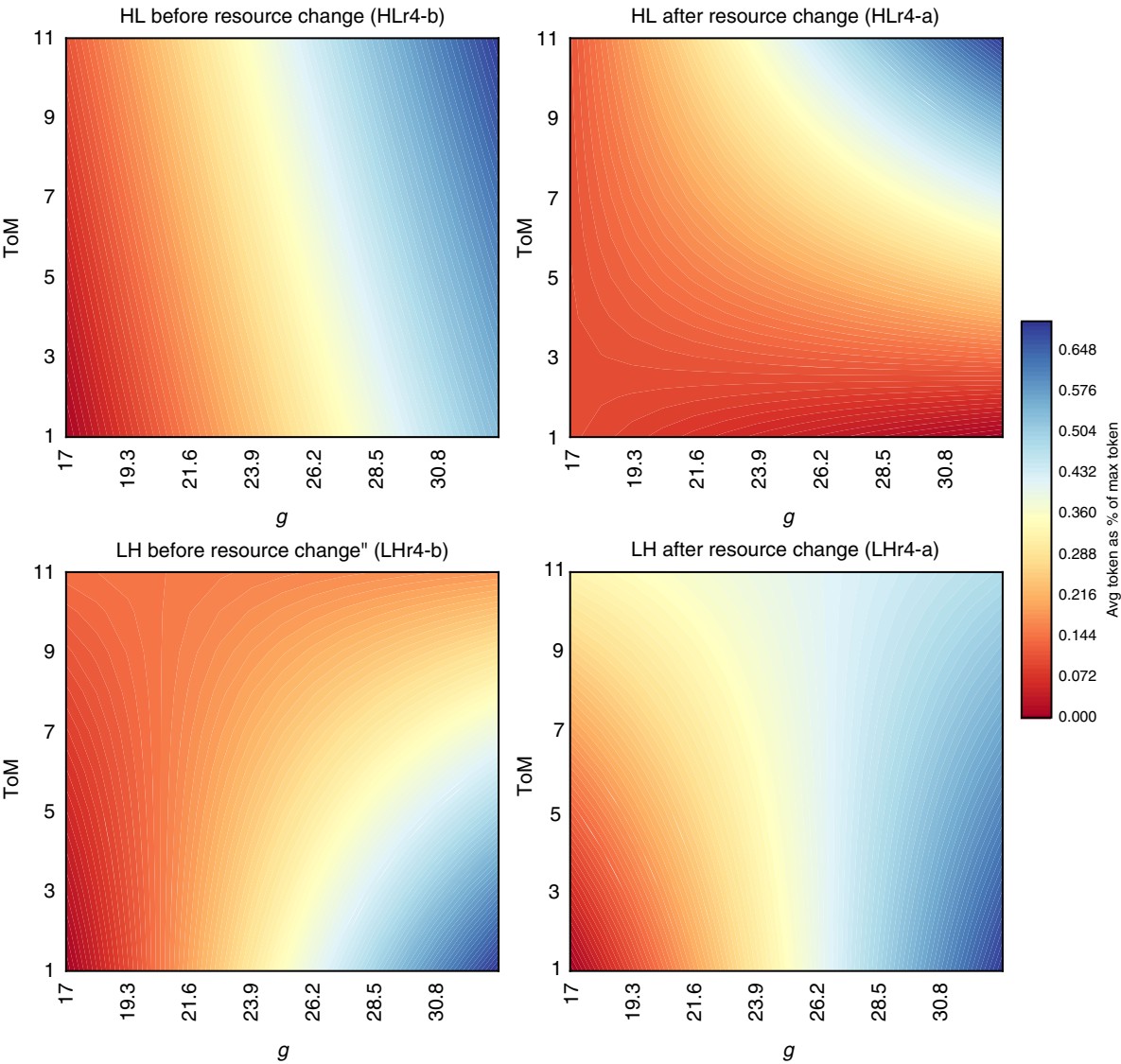

**Fig. 4** Effects of *g* and ToM and change on average tokens collected as % of maximum tokens available. Blue color group harvested less, HL high-to-low resource growth treatment, LH low-to-high resource growth treatment, -b marginal effect when change = 0 (before change), -a marginal effects when change = 1 (after change). The different figure panels represent the marginal effects of *g* and ToM for models *Hr4* and *LHr4* portrayed in Supplementary Table 5

informational diversity may decline with prolonged contact between team members[50,51,56,57]. Our approach contributes to sorting out why the effects of diversity are contextual by considering *g* and ToM as analogous to functional traits that provide the cognitive infrastructure for individuals and groups to adapt to social and ecological change.

Overall, our results illustrate that round-to-round, groups with a high *g* but low ToM spend the most time staring at a blank screen and put the most pressure on the resource after an ecological change (see Figs. 2 and 3). However, only when conditions deteriorate do groups with high *g* and high ToM really reap the benefits of functional diversity, in terms of overall tokens harvested (see Fig. 4). Over the first three rounds, groups with high *g* and low ToM understand how to optimize token harvest and work to maximize their profit. This behavior also implies pushing the resource closer to its breaking point and risks a collapse. This risk materializes more often round-to-round among high *g* and low ToM groups. At the same time, groups with high ToM and low *g* are also more likely to increase pressure on resources as shown in Fig. 3, especially when the

growth rate of resources declines (HL treatment); however they are, we speculate, better able to re-negotiate and avoid the resource collapse than high *g*, low ToM groups as depicted in Fig. 2.

Based on the experimental evidence presented here, we reason that groups with high *g* and high ToM manage a common pool resource more sustainably than groups with high *g* or ToM alone, especially in the case of a negative change to the resource (HL treatments). In fact, our results indicate that groups either high in *g* or high in ToM alone harvest common pool resources less sustainably than groups with high values of *g* and ToM. This result is most robust in the HL treatment (see HL panels in Figs. 2 and 3, Supplementary Tables 2 and 3). In our LH treatment, the effects are the same, but not significant with respect to speed of resource depletion (see Supplementary Table 2). In the LH treatment, the history of success or failure in the previous round is the main variable with a significant effect on the sustainable harvest of resources. Thus, as anticipated, cognitive abilities seem less salient for managing the resource sustainably when the resource conditions improve.

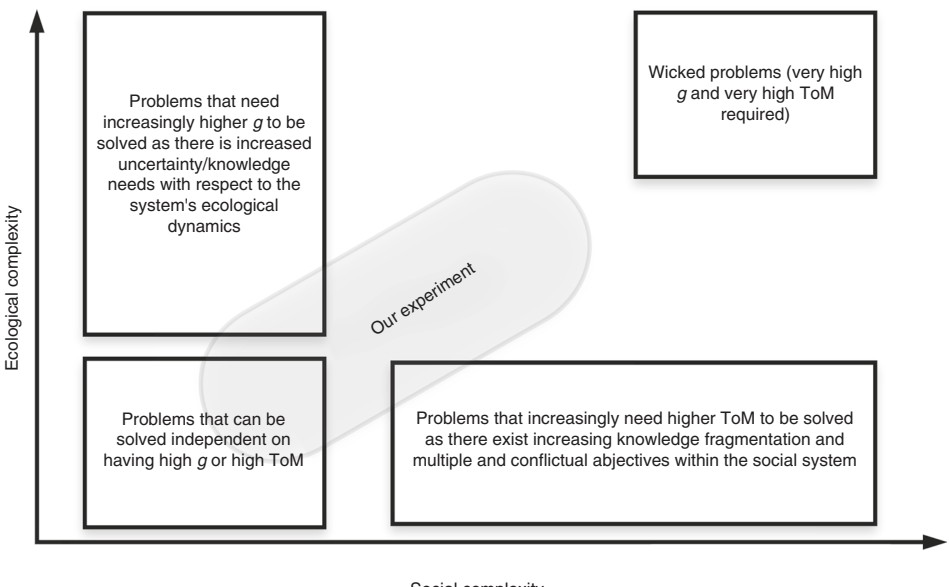

**Fig. 5** Relation between ecological complexity, social complexity (including contentiousness of the problem) and relative importance of *g* and ToM

Taken together, the experimental results presented here hint at the possibility that a deficit in either *g* or ToM leads to less beneficial group outcomes (sustained resource use), and, especially where *g* is high and ToM low, more selfish behaviors that maximize an individual's short-run harvest. The patterns observed fit experimental data beyond our study. For example, in time pressure experiments, which reduce cognitive function (*g*), individuals cooperate much more in public goods games and one shot prisoner's dilemmas[58]. Further, in a meta analysis of 76 studies, Imuta and colleagues find a positive and statistically significant association between ToM and pro-social behavior among children 2–12 years old[59].

Our results are also consistent with recent literature in psychology suggesting that ToM (but not *g*) is the key individual level factor predicting how well groups perform on a battery of tasks (group intelligence)[39]. Groups with high *g* and low ToM harvest faster and push the resource system closer to its breaking point following a change in resource growth rate. This does not occur where both *g* and ToM are high, perhaps signaling the emergence of a more well developed group intelligence. Similarly, Gonzalez-Forero and Gardner use a first principles model of metabolic costs to illustrate that a combination of two types of games, individuals vs. nature (selecting for high *g*) and groups vs. nature (selecting for higher ToM or more social learning) accounts for 90% of variation in brain size and life history of our species[60]. Our results are consistent with the idea that both individual level intelligence and functioning as social groups are important for humans to solve social-ecological challenges.

We suspect that the salience of a functional diversity of cognitive abilities for sustainably managing resources depends on a system's social and ecological complexity. In fact, we can classify any social-ecological governance challenge along two axis, one related to the degree of social complexity and one to the degree of the ecological complexity (see also ref. [61]). Social complexity refers to the number of different objectives, the degree of contentious objectives, the degree of cooperation or conflict, and the degree of knowledge fragmentation. Ecological complexity refers to the diversity, non-linearity, and predictability of processes within the ecological system. Our experimental treatments fall between a simple and a wicked problem (see Fig. 5). The low-to-

high treatment effectively represents a simpler problem in which intelligence is much less salient and reciprocity takes over as the main driver of harvest dynamics. We propose that the salience and benefits of functionally diverse and high competence cognitive abilities increases as a problem increases in both social and ecological complexity. However, to better assess the relationship between functional diversity of cognitive abilities and social and ecological complexity, more studies are needed. Future experimental studies could increase the ecological complexity by changing the resource dynamics (different threshold of regrowth, introducing high non-linear growth functions, giving different types of signal or cues etc.) as well as the social complexity by, for example, priming participants differently, and/or allowing for different compensations dependent on different objectives for each participant (or for groups of participants).

To summarize, collective action is essential for social groups to solve complex ecological problems. While it is known that institutions –rules and norms– have a key role in increasing sustainability of common pool resources, it is individuals that lie at the base of institutions. Individuals make and change rules and norms as environments change. Yet, there have been few studies that investigate the role of cognitive abilities in promoting collective action in changing ecological settings. Cognitive abilities underlay the ability of individuals to build models of their social and ecological circumstances and, thus, make decisions about how to collectively manage resources. In this paper we have begun to disentangle the relationship between individual cognitive abilities (ToM and *g*) and the ability of groups to act collectively to manage common pool resources sustainably in a changing environment. Consistent with the functional intelligences proposition, results from two experimental treatments illustrate how groups with both high *g* and ToM outperform groups with high *g* or ToM alone, especially when faced with negative environmental change.

Hence, in our experimental setting, a diversity and high competency of different intelligences—*g* and ToM—is key to sustainably manage resources. Just as a functional diversity of traits increases and maintains the efficiency and stability of ecosystems, a functional diversity of intelligence increases the ability of groups to sustainably harvest resources, especially in degrading environments.

## Methods

**The experiment in a nutshell**. The common pool resource system consists of a spatially dispersed resource (tokens) that grows according to a density dependent function (see Supplementary Method 5). The growth of the resource is not known by the participants. Each participant in our experiment was paid $0.02 per unit of resource harvested (tokens). Thus, individuals constantly faced the temptation to harvest as many tokens as they could, as quickly as they could, to maximize their revenue in the short-run. However, so doing always has a community cost in the experiment: the resource base may be depleted quickly. In each experimental treatment, groups of four anonymous individuals harvest tokens for 6 rounds (180 s each) on a 20×20 grid (see Supplementary Method 5). In treatment #1 (high resource-to-low resource growth—HL), we evaluated the effect of a negative change in the growth rate of the resource base on the ability of groups to collectively harvest tokens. In this treatment, individuals harvest tokens in rounds 1–3 with a high re-growth rate, and in rounds 4–6 with a low regrowth rate. In treatment #2 (low resource-to-high resource growth—LH), we evaluated the effect of a positive change in the growth rate of the resource base on the ability of groups to collectively harvest tokens. In this treatment, individuals harvest tokens in rounds 1–3 with a low regrowth rate, and in rounds 4–6 with a high regrowth rate. Both negative and positive change relate to a halving or doubling of the resource growth rate. In both experimental treatments, individuals were allowed to communicate before each round of the game but were never informed about the change in the growth rate of the resource base nor how many rounds they would play. See Supplementary Method 5, 6 and 7 for the experimental design, recruitment and protocol. This study complied with with all relevant ethical regulations for work with human participants, and informed consent was obtained by each participant. This study was approved by the Institutional Review Board (IRB) at Utah State University (protocol # 7664) and at the University of Texas at San Antonio (document # HRP-522, IRB number 16–256).

**Measuring governance of the resource**. The first performance measure is the Time per round that a group stares at a blank screen because they have collapsed their resource base. Time relates to the ability of groups to avoid collapse of resources before the time of a round comes to an end. We rescale time to the [0,1] interval, where 0 = tokens harvested until the end of a round and 1 = indicates an immediate resource collapse. The second performance measure is the change in performance between the first three[1,3] and the second three rounds[4,6] based on the maximum theoretical average tokens collected: $\Delta T$, where $\Delta T = \mathrm{Avg}T_{1,3} - \mathrm{Avg}T_{4,6}$. $\mathrm{Avg}T_{1,3}$ is the average number of tokens collected as a % of maximum possible tokens in the first three rounds (55 token per individual in case of high growth rate, and 36.25 tokens per individual per round in case of low growth rate), and $\mathrm{Avg}T_{4,6}$ is the average number of tokens collected in the second three rounds (calculated as described for the first three). We also employ avg$T$ on its own in order to assess group performance as the number of tokens collected by groups each round (see Supplementary Method 1, Supplementary Table 1 and Supplementary Figs. 4–6 for more details on how $\Delta T$ and avg$T$ were calculated). avg$T$ thus represents the average tokens harvested (as % of maximum possible tokens) of a group in rounds 1–3 and in rounds 4–6. All three dependent variables measure different aspects of group performance with respect to resources. Groups that increase harvest pressure after changes (higher $\Delta T$ are possibly not identifying local resource dynamics, and/or are not able to re-negotiate resource allocation and harvest, hence leading to a higher probability of resources collapsing (higher Time) and overall fewer tokens collected (lower avg$T$). On the other hand, when environmental conditions improve, we would not expect a strong relationship between increased harvest pressure and resource depletion nor overall tokens collected.

**Measuring g, ToM, and other independent variables**. To measure $g$, participants were asked to report their official ACT/SAT scores. ACT/SAT scores correlate highly with IQ scores and other measures of $g$ (corrected $r = 0.86$[62–64],), which drives the predictive validity of cognitive tests[13,270–301]. We used equivalence tables from the College Board 2016 in order to transform SAT scores into ACT scores[65]. As a proxy for group $g$ we averaged such scores at the group level. To measure ToM, each participant in our experiments completed a short story test designed to measure social reasoning (the SST)[66]—see also Supplementary Note 1 and Supplementary Method 8. The SST requires reasoning about the mental states of characters in a short story[66] and measures social-cognitive theory of mind[67]. Social cognitive ToM measures the ability to infer others' intentions and plan potential courses of action. As a proxy for group ToM we used the minimum ToM score within a group, following the saying that one "low ToM" can have detrimental effects on the overall group dynamics by increasing conflict and reducing communication effectiveness. For more information on the variables used and their descriptive statistics (see Supplementary Note 2 and Supplementary Figs. 1–3.)

The use of average $g$ and minimum ToM to estimate group values of $g$ and ToM follows from the functional diversity of tasks that the two different cognitive abilities allow individuals and groups to perform. $g$ is key to understand the system, and understand changes affecting the system, hence it is important that skills between participants can be averaged in order to increase the likelihood of a positive team outcome[29]. ToM is key for smooth social interactions. To maintain smooth social interactions all members of the group need to communicate and diffuse tension, which depends on understanding others at a minimal level. One

individual with very low ToM has the ability to jeopardize the effectiveness of the whole group[38]; hence, tasks that are related to ToM are more akin to conjunctive tasks, and the minimum level should determine the overall group ability to perform a specific task[29].

Finally, we measure ethnic and religious diversity as $-\Sigma(C_i * \log(C_i))$ where $C_i = N_i/N$ represents the fraction of individuals of religion or ethnicity $i$ within a group of four. We assess gender as the % of males within a group. We measure trust via survey questions (see Supplementary Method 9) and chat volume as the number of messages exchanged within a group per round.

**Statistical models**. To assess the effect of $g$ and ToM on $\Delta T$ we employ a simple linear regression model with interaction between $g$ and ToM. The difference in communication between the first three and the second three rounds is based on: $\Delta$Chat = Avg Chat$_{1,3}$ − Avg Chat$_{4,6}$.

To assess the effect of $g$ and ToM on Time we employ a general linear model. Following Papke and Wooldrige[68] the model was estimated via quasi-maximum likelihood as suggested by Gourieroux[69]. Given the relationship between cognitive abilities (as shown in[70]), we assess the following models for both treatments separately (low-to-high, LH, and high-to-low HL resource regrowth rates, representing, respectively a positive (LH) and a negative (HL) change in the environment)—see Supplementary Method 2, 3 and 4, and Supplementary Tables 2–9 for model results:

$$E|\mathrm{Time}_t = \beta_0 + \beta_1 * g + \beta_2 * \mathrm{ToM} + \beta_3 * g * \mathrm{ToM} + \overrightarrow{\beta_i * \mathrm{x}} + \mathrm{R}_t + \varepsilon \quad (1)$$

$$\Delta T = \beta_0 + \beta_1 * g + \beta_2 * \mathrm{ToM} + \beta_3 * g * \mathrm{ToM} + \overrightarrow{\beta_i * \mathrm{x}} + \varepsilon \quad (2)$$

and

$$\mathrm{avg}_t = \beta_0 + \beta_1 * g + \beta_2 * \mathrm{ToM} + \beta_3 * \mathrm{change} + \beta_4 * g * \mathrm{ToM} + \beta_5 * g * \mathrm{change} + \beta_6 * \mathrm{ToM} * \mathrm{change} + \beta_7 * g * \mathrm{ToM} * \mathrm{change} + \varepsilon \quad (3)$$

where $\overrightarrow{\beta_i * \mathrm{x}}$ is the vector representing additional variables: $\Delta$Chat for Eq. (1), and Chat volume for Eq. (2), ethnic diversity, religious diversity, and gender. $R =$ round groups are playing (only used in Eq. (2)), and $\varepsilon$ represents the error term.

The GLM is estimated by rescaling the dependent variable Time in the [0,1] interval, where both 0 and 1 have a theoretical positive probability to be an actual outcome. Such rescaling allows us to estimate the expected time that each group is able to use to harvest tokens: $E|\mathrm{Time} = \mathrm{f}(\overrightarrow{\beta x})$ is estimated via quasi-maximum likelihood, where $E|\mathrm{Time}$ is the expected harvesting time per round, and $\overrightarrow{\beta x}$ is a vector of model variables including the interaction between $g$ and ToM, with the addition of the following control variables (inserted one by one, see Supplementary Table 2): communication volume (difference between the first three rounds and the second three rounds), ethnic diversity, religious diversity and gender. $f(\cdot)$ is the logistic function: $f(g) = e^g/(1 + e^g)$.

**Ethical approval**. This study was approved by the Institutional Review Board (IRB) at Utah State University (protocol # 7664) and at the University of Texas at San Antonio (document # HRP-522, IRB number 16-256).

## Data availability

All data and codes used to generate figures and tables presented here and in the supplementary information are available upon request to the corresponding author.

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

## Acknowledgements

The authors have been supported by NSF Grant SMA-1620457.

## Author contributions

J.A.B., J.F., and T.R.C., designed the research; J.A.B. and J.F. analyzed the data; J.A.B., J.F., T.R.C., T.T.N., D.H., K.E.E., S.N., and D.P. performed the research, J.A.B., J.F., T.R.C., T. T.N., D.H., K.E.E., S.N., H.J.F.D., and D.P. wrote the paper

## Additional information

**Competing interests:** The authors declare no competing interests.

