## [Peer Review File · Nature Communications]

Reviewers' comments:

Reviewer #1 (Remarks to the Author):

This study uses the real-time common-pool resource game created by Janssen et al. (2009) to test the "Functional Intelligence Proposition" (FIP). The novelty introduced by the study authors is in changing the rate of resource regrowth across rounds to serve as a proxy for exogenous shifts faced by resource harvesters mimicking climate change or other rapid environmental changes. The authors separately gather information on subjects' general intelligence (g) and social intelligence ("Theory of Mind" or ToM). The hypothesis of the study is that higher levels of g should lead to efficiency-gains in the short run, but if the group has a low ToM, groups with high or low g will not be as effective as obtaining long-run sustainability (re: efficiency over time) as groups with high ToM. Thus, the authors expect that socially-conscious groups, proxied by high ToM scores, will be better at adapting to shifts in harvesting rates across rounds than individually-conscious groups.

Overall, this study is well-executed and seems to be a solid contribution to our understanding of the conditions under which common-pool resource harvesting will be (un)successful at achieving long enduring, sustainable, resource harvests. The use of the experimental setting is appropriate and the statistical analyses, while rudimentary, seem appropriate to the study under investigation. In general, it is straight-forward in presentation and I have only a couple of major concerns that, in my reading, would prohibit publication in its current form.

Major concerns

The authors approach the topic from the vantage of psychology, so I am not familiar with FIP as a theoretical concept and can't speak to its importance in that field. However, what is sorely lacking from the manuscript right now is any discussion of collective action dilemmas, which are really the core of the study of common-pool resources. The authors do briefly note the literature on heterogeneity in common-pool resource dilemmas, but one paragraph linking this literature—which really studies collective action and not cognitive capabilities—with psychology would be a useful bridge. Particularly in how g is expected to influence collective action.

In some ways, isn't a high g simply saying "the group is filled with rational actors", in which case all this experiment boils down to is that groups filled with rational actors are less likely to overcome collective action dilemmas than groups filled with actors with other-regarding preferences? Bruno Frey has written on this in experiments.

Minor concerns

Lines 33-35—Definition of common-pool resource includes both rivalry/subtractibility (as noted), but also non-excludability, which is really the crux of the collective action dilemma

Perhaps the hypothesis/expectations would be easier to interpret with a simple 2x2 figure that lists high/low g on one axis and high/low ToM on the other axis with each cell filled with expectations on Time & change-in-efficiency (the two dependent variables).

What is 'SI' in the interaction equations?

Reviewer #3 (Remarks to the Author):

The paper tests the hypothesis that two types of intelligence (general and social) have an interactive effect on a group's governance of common pool resources. The hypothesis is tested in an experimental setting, and some interesting findings are presented. Overall, the experiment is

well-designed, however the development of the paper could be significantly improved. Further, the conclusions that can be drawn from this study should be much more carefully presented, in order to avoid overclaiming. I hope the following comments will be helpful in developing the paper further:

Theory and hypotheses development

1. The current setup makes an unqualified assumption about the orthogonality of g and ToM, which needs to be thought about deeply, and defended.

For instance: Is ToM a prerequisite for g? Is there a latent variable that predicts both g and ToM? Are g and ToM orthogonal? In the field of animal cognition, there is rich discussion of theory of mind and intelligence, that I suspect could be quite useful (Wynne, 2001, Chapter 1). Further, while both forms of intelligence have been put in the cognitive umbrella, the group processes that are thought to be impacted are cognitive (for g) and social (for ToM); clear specification and development of this cognitive-social link from will be useful.

2. There is a rich literature in psychology and organizational behavior that deals with the topic of group diversity; I point to some reviews that might help the authors (Bell, 2007; Mannix & Neale, 2005). In the current manuscript, there are several holes in the theoretical development and justification:

a. How are the authors thinking about diversity in this context? Is the highest level of functional diversity in a group achieved when there are both high levels of g and ToM, and hence both of these are resources that the group can use, or is the highest level of functional diversity achieved when there are high levels of one but low levels of another? A useful citation is Harrison & Klein, 2007 (Harrison & Klein, 2007).

b. Currently, the paper adopts a simplistic approach to aggregation to the group level, without consideration of the complex approaches and their theoretical justification. In small groups and teams research, much thought has been given to aggregating data to the group level. The following citations should be useful in justifying why the approach of taking the average was used (Barrick, Stewart, Neubert, & Mount, 1998; LePine, Buckman, Crawford, & Methot, 2011; Meslec, Aggarwal, & Curşeu, 2016).

c. There is work on collective intelligence or intelligence that emerges at the group level and cannot be attributed solely to the individual team members (and hence is different from the individual-aggregation level). While the work on collective intelligence has been cited, its implications have not been considered in the theoretical development of the paper. (Meslec et al., 2016; Woolley, Aggarwal, & Malone, 2015)

3. Theoretical loose ends

a. While from the Abstract and the Discussion and Conclusion sections of the paper, the "long-term" versus "short-term" aspect of utilization of resources seems to be important, it is not mentioned nor developed in the front-end of the paper.

b. Citations should be provided to qualify the claim that groups with higher g should be better at adapting to changes in their resource base as they are better at detecting changes in a system (Pages 3-4). Additionally, given the importance of change to the study (the experimental manipulation), there should be more about change and its association with group composition in the theoretical development.

4. Contribution: As pointed out on Page 3, it seems that the FIP states that general intelligence and social intelligence are both capabilities that are critical for social groups to govern resources. Hence, is the contribution of this paper the interactive effect and not just the additive effect of the two types of intelligence? This should be presented clearly.

Study

1. Was the sample size 54 groups? And 27 in each treatment? This information should be explicitly provided

2. The authors would need to defend that they had enough power to run six regression models per treatment with several control variables.

3. Currently, while I understand that the authors computed two dependent variables, the

theoretical link between the two is unclear. Are they indicative of a similar theoretical construct? If not, could the authors build more up front in order to establish two distinct dependent variables.

4. From the Results section, the data showed a synergistic interactive effect in only the high-to-low condition, for the first dependent variable (Time). However, there is a compensatory interactive effect for both conditions with the second dependent variable (Delta T). These findings need to be reconciled and elaborated with much more cohesion.

5. Suggestion: When computing the second dependent variable, instead of using the average of the three times before and after the treatment and then calculating the difference, could the authors calculate the slope using OLS regression for the three times before and the three times after and calculate the difference in the slopes? Would that measure capture the richness of the longitudinal data in a richer way? Does it yield similar results? The first round's scores will still need to be controlled for.

Discussion and Conclusions

1. The Discussion section needs to be much more crisp and attentive to the results; for example on Page 14 (and 18), the authors write "we reason.... and our analysis is consistent with FIP." The authors only found a synergistic interactive effect for one of the treatment conditions and only for one of the dependent variables (Time). The authors need to be very precise in the deductions they make from the results to avoid overclaiming.

2. As pointed above, much more can be done to tie together the two dependent variables and establish what theoretical variable they are capturing; further the short-term, long-term distinction, while appealing, is not currently developed adequately, and several concerns are raised in the reader's mind. For example, can a short-long term claim really be made given the very short time scale of the experiment (where each round was 180 seconds).

3. External validity: the authors need to convince the audience of the external validity of this research question.

Suggested references:

Barrick, M. R., Stewart, G. L., Neubert, M. J., & Mount, M. K. (1998). Relating member ability and personality to work-team processes and team effectiveness. *Journal of Applied Psychology, 83*(3), 377.

Bell, S. T. (2007). Deep-level composition variables as predictors of team performance: a meta-analysis. *Journal of Applied Psychology, 92*(3), 595.

Harrison, D. A., & Klein, K. J. (2007). What's the difference? Diversity constructs as separation, variety, or disparity in organizations. *Academy of Management Review, 32*(4), 1199–1228.

LePine, J. A., Buckman, B. R., Crawford, E. R., & Methot, J. R. (2011). A review of research on personality in teams: Accounting for pathways spanning levels of theory and analysis. *Human Resource Management Review, 21*(4), 311–330.

Mannix, E., & Neale, M. A. (2005). What differences make a difference? The promise and reality of diverse teams in organizations. *Psychological Science in the Public Interest, 6*(2), 31–55.

Meslec, M. N., Aggarwal, I., & Curşeu, P. L. (2016). The insensitive ruins it all: Compositional and compilational influences of social sensitivity on collective intelligence in groups. *Frontiers in Psychology, 7*, 676.

Woolley, A. W., Aggarwal, I., & Malone, T. W. (2015). Collective Intelligence and Group Performance. *Current Directions in Psychological Science, 24*(6), 420–424.

Wynne, C., D. L. (2001). *Animal Cognition: The Mental Lives of Animals*. Palgrave Publishers.

Reviewer #4 (Remarks to the Author):

This is a very interesting paper. The research question is strong and interesting, and it is timely given this ever important topic (sustainability). Functional diversity is an important ingredient in maintaining sustainability. The authors have done a very good job of isolating and measuring the two variables of interest (general intelligence and social intelligence). I thought that using the SAT/ACT scores from participants to estimate general intelligence was particularly well done.

Furthermore, their statistical analysis is good and the robustness checks strengthen their results, though I have a few questions and suggestions below.

MAJOR POINTS:

1) I looked at your models carefully and liked the robustness checks that you conducted: controlling for chat, trust, religious and ethnic diversity, and gender are all useful in providing good evidence of your phenomenon. However, one IV I couldn't find (but seems rather important) is an indicator variable for the change of growth rate within each treatment. That is, I'd expect a "\delta growth" indicator variable and a (three-way) interaction with ToM and g in your models. After all, if I understand your argument correctly, you are proposing that ecological change (i.e. the missing indicator variable) has differential effects depending on the ToM and g of your groups. Correct?

In short, I think the following ought to be clarified: what is the effect of the growth rate change (as described above) and, equally important, what would be your predictions for such a three-way interaction? I'd spend some more time/space in the introduction clarifying what you expect to see in your different treatments and why.

2) My second main question is around the choice of metrics in our main models. You chose to focus on the smallest social intelligence in each group ($\min(\text{ToM})$) and the average general intelligence in each group ($\text{avg}(g)$) in your main model (Tables 2 and 3 in the SI). It seems like an arbitrary choice to use "min" for one metric and "avg" for the other, and it was not motivated in the introduction/theory section. To be honest, I'm not familiar with any literature that argues convincingly that either min or avg are the appropriate measures of general/social intelligence in groups to use, but perhaps there is – it seems important to look into this.

On a related note, I appreciate that the authors included other variants of the model specification, e.g. $\text{avg}(g) * \text{avg}(\text{ToM})$ and $\min(g) * \text{avg}(\text{ToM})$ and $\min(g) * \min(\text{ToM})$ in the SI (Tables 4-7). And while not all models show a significant interaction, the evidence in aggregated across the tables seems to be consistent with the predicted significant interaction, so that makes sense. That said, I'd push the authors to think harder about which metrics (avg or min) should be used and why, and use that as the main table to refer to in the main text. Unless there is a good reason for it, I'd suggest sticking with either min for both or avg for both.

As a purely empirical, post-hoc observation to stimulate your thinking, I found it intriguing that the models using $\min(g)$ and $\min(\text{ToM})$ were consistently and significantly in the predicted direction – one way to read this (again, caution because post-hoc interpretation) is that the "weakest link" in the group matters: the lowest general and social intelligence is a good predictor of how well the group does.

3) Lastly, I understand your arguments around the synergetic effects of $\text{ToM} * g$. However, I couldn't quite understand the main effects of those variables. That is, at the moment there are no regression models with just ToM and g as main effects (without the interaction). That would be a first important addition.

Second, I couldn't quite make sense of the single effects of ToM and g in your existing models. E.g. looking at Table 3, it makes sense that $\text{ToM} * g$ is significant and negative (i.e. reducing the amount overharvested). But why is the single effect of ToM (0.065***) and g (0.023***) positive? I think it may make sense to plot the actual data on a ToM vs. g heatmap grid, and see what the effects in each corner are like when independently varying just one variable, and explaining this to the reader (perhaps the marginal plots are already doing this, but I couldn't quite follow it – I felt the labels and titles in the graphs were not very easy to understand on their own).

Perhaps to make it easier for readers to read these additional results, I'd suggest you create new columns with just the main effects (without interaction), a column with single and interaction effects (like your column 1 in your existing models), and a column with all your covariates added in simultaneously (akin to your column 6).

MINOR POINTS:

* I liked your Figure 3 but felt that it came way too late in the text. I'd move this upwards into the introduction to give people a better sense of the magnitude of the problem you are trying to tackle. By the way, incidentally, I would encourage you to use the same figure to add in the number of groups (N=...) that you have that are low g/low ToM, low g/high ToM, high g/low ToM and high g/high ToM, so that readers can immediately see that you cover the whole spectrum of interest in your experiment/analysis.

* I thought your argument to include round number in the analysis was sensible. Why did you only include it in the models predicting time spent thinking, but not in the ΔT models? I'd add them in the latter too.

* Personally, I think that the evidence you present on ΔT was more convincing than the time spent thinking about the resource. This is because some literature (e.g. Evans, Dillon & Rand, 2015) would suggest that the latter (decision time) does not indicate the mental state of individuals (not sure there's evidence of teams) but rather indicates the extent to which an individual faces decision conflict. So to the extent that you want to demonstrate that teams exercise restraint in their self-interested choices, I'd suggest that ΔT is a better proxy than decision time.

* Clarification question of how you define "time spent looking at the screen" (one of your outcome variables): given that in low-growth environments a resource takes longer to re-grow, I assume you normalize the "time spent" variable by the time it takes a resource to grow to full capacity, correct? (Otherwise, it seems you'd bias the low-to-high treatment against the other due to slow start/learning effects in earlier rounds.)

* At the moment you do not compare the two treatments (HL vs. LH) in your models (i.e. using treatment assignment as an IV), so I'd be careful with the language you use to describe the results. E.g. "Figure 1 illustrates that groups with both high g and high ToM are less likely to collapse their resource, especially in the high-to-low treatment, before a round is finished."  The result you refer to is actually only true in the high-to-low treatment, not "especially" – this may be a minor point but I'd be precise in your language based given your models.

* How were people treated in the analysis who were familiar with the SST? If those familiar are excluded from the analysis, what happens if they are included?

* A recently published paper in Nature might be of relevance and should probably be included in the discussion of cognitive (intellectual) and social drivers of cooperation: González-Forero, M., & Gardner, A. (2018). Inference of ecological and social drivers of human brain-size evolution. *Nature*, 557(7706), 554.

Reviewers' remarks to authors and author replies

Reviewer #1 (Remarks to the Author):

This study uses the real-time common-pool resource game created by Janssen et al. (2009) to test the "Functional Intelligence Proposition" (FIP). The novelty introduced by the study authors is in changing the rate of resource regrowth across rounds to serve as a proxy for exogenous shifts faced by resource harvesters mimicking climate change or other rapid environmental changes. The authors separately gather information on subjects' general intelligence (g) and social intelligence ("Theory of Mind" or ToM). The hypothesis of the study is that higher levels of g should lead to efficiency-gains in the short run, but if the group has a low ToM, groups with high or low g will not be as effective as obtaining long-run sustainability (re: efficiency over time) as groups with high ToM. Thus, the authors expect that socially-conscious groups, proxied by high ToM scores, will be better at adapting to shifts in harvesting rates across rounds than individually-conscious groups.

Overall, this study is well-executed and seems to be a solid contribution to our understanding of the conditions under which common-pool resource harvesting will be (un)successful at achieving long enduring, sustainable, resource harvests. The use of the experimental setting is appropriate and the statistical analyses, while rudimentary, seem appropriate to the study under investigation. In general, it is straight-forward in presentation and I have only a couple of major concerns that, in my reading, would prohibit publication in its current form.

REPLY:

We would like to thank the reviewer for their comments. We believe that the reviewer's suggestion to better integrate the cognitive and common pool resource literature has improved the paper.

REVIEWER #1 REMARK:

Major concerns

The authors approach the topic from the vantage of psychology, so I am not familiar with FIP as a theoretical concept and can't speak to its importance in that field. However, what is sorely lacking from the manuscript right now is any discussion of collective action dilemmas, which are really the core of the study of common-pool resources. The authors do briefly note the literature on heterogeneity in common-pool resource dilemmas, but one paragraph linking this literature—which really studies collective action and not cognitive capabilities—with psychology would be a useful bridge. Particularly in how g is expected to influence collective action.

In some ways, isn't a high g simply saying "the group is filled with rational actors", in which case all this experiment boils down to is that groups filled with rational actors are less likely to overcome collective action dilemmas than groups filled with actors with other-regarding preferences? Bruno Frey has written on this in experiments.

REPLY:

Indeed we agree with the reviewer. And for this reason we have rewritten the introduction and linked cognitive abilities with common pool resources and collective action dilemmas more explicitly. In fact there is a parallelism between selfishness and individuals with high g and low ToM, as well as a link between g and ToM and the ability of groups to devise rules that match local conditions, monitor resource, establish graduated sanctions, minimize conflict and devise rules that take proportionality of investment and extraction into account. In other words, g and ToM, given the functions they accomplish cognitively, affect how groups are able to implement and adopt Ostrom institutional design principles, which, in turn, increase the likelihood of successful common pool resource management.

REVIEWER #1 REMARK:

Minor concerns

Lines 33-35—Definition of common-pool resource includes both rivalry/subtractibility (as noted), but also non-excludability, which is really the crux of the collective action dilemma

REPLY:

Indeed the reviewer is right. We have amended the text and now make the non-excludability and rivalrousness of goods explicit (see lines Introduction)

REVIEWER #1 REMARK:

Perhaps the hypothesis/expectations would be easier to interpret with a simple 2x2 figure that lists high/low g on one axis and high/low ToM on the other axis with each cell filled with expectations on Time & change-in-efficiency (the two dependent variables).

REPLY:

We thank the reviewer for this suggestion, and we have added Figure 1 in order to clarify our hypothesis and better relate g and ToM to resource depletion and harvest pressure pre-post harvest change. We do think that Figure 1 will aid readers understanding of the proposed relationship between cognitive abilities (g and ToM specifically) and the ability to manage common pool resources sustainably.

REVIEWER #1 REMARK:

What is 'SI' in the interaction equations?

REPLY:

We thank the reviewer for this comment. We have changed SI to ToM (SI referred to social-intelligence in a previous version). Theory of Mind is more appropriate as this is what we are eliciting via the test proposed by Dodell-Feder et al. 2013).

Reviewer #3 (Remarks to the Author):

The paper tests the hypothesis that two types of intelligence (general and social) have an interactive effect on a group's governance of common pool resources. The hypothesis is tested in an experimental setting, and some interesting findings are presented. Overall, the experiment is well-designed, however the development of the paper could be significantly improved. Further, the conclusions that can be drawn from this study should be much more carefully presented, in order to avoid overclaiming. I hope the following comments will be helpful in developing the paper further:

REPLY:

We would like to thank the reviewer. The reviewer makes incisive and sophisticated comments and has challenged us to think deeply about the theory and interpretation of results. We think, as a result, the paper is much stronger and higher quality.

REVIEWER #3 REMARK:

Theory and hypotheses development

1. The current setup makes an unqualified assumption about the orthogonality of g and ToM, which needs to be thought about deeply, and defended.

For instance: Is ToM a prerequisite for g? Is there a latent variable that predicts both g and ToM? Are g and ToM orthogonal? In the field of animal cognition, there is rich discussion of theory of mind and intelligence, that I suspect could be quite useful (Wynne, 2001, Chapter 1). Further, while both forms of intelligence have been put in the cognitive umbrella, the group processes that are thought to be impacted are cognitive (for g) and social (for ToM); clear specification and development of this cognitive-social link from will be useful.

REPLY:

The reviewer raises three important issues. One issue is whether g and ToM really do serve distinct information processing function (are orthogonal). We would note that complete orthogonality is not necessary for these two cognitive abilities to serve distinct functions, though some degree of orthogonality is necessary.

In this revised version of the manuscript, we now justify the underlying premise that g and ToM serve different information processing roles or capacities. Three lines of evidence are cited.

1. Neurological studies of brain region activation.
2. Evidence that cause—effect detection and reasoning are widespread among animals, but ToM is not, and the most convincing ToM evidence comes from highly social species with complex communication systems.
3. Weak correlations between Social Cognitive ToM and aspects of g among human subjects

We have added a paragraph to the ``theory section of the article and we have presented correlations between g and social cognitive ToM in the supporting information.

A second issue is whether ``ToM is a prerequisite for g.” This is beyond the scope of our paper. However, as a really interesting point of discussion, the comparative psychology that the reviewer pointed us toward would seem to suggest, potentially, the opposite. There is a broad selective advantage for some level of g competency (detecting association, making inferences from chains of association). The rarity of ToM would suggest that this ability only provides a selective advantage, perhaps, when a species depends highly on ToM when issues and problems must be solved in groups and where understanding the underlying system needs to be communicated and solutions agreed upon. Only in these relatively rare situations (in nature) does selection favor groups whose individuals that are able to employ complex mental ability such as modelling others intentions. This would imply that g is, in a sense, prior to, ToM.

A final issue is the social-cognitive link. We have now attempted to clearly specify that ToM is an individual level attribute that affects communication, conflict, and feeling cheated. These are social processes. G affects the ability to represent resource dynamics and this is an individual cognitive process, though mental representations may be modified through communication. We have attempted to link the two through the Nobel Prize study of E. Ostrom (1990) on the governance of common pool resources. She specifies eight design principles for governing common pool resources. Two of the most important are clear boundary identification and the matching of rules to resources. Both of these rules require cognitive representations (understanding) of the resource system. Three other important rules include monitoring others’ behaviors, inclusive decision making and rules for graduated sanctions. All of these invoke social processes that ToM is critical to make more efficient. It is more efficient to monitor others if one can model their mental state. Inclusive decisions are more efficient with higher ToM as individuals are more willing to take another’s position and compromise. Similarly, higher ToM means better communication about why rules were violated and more understanding of the reasons, which wards off excessive retribution.

We now discuss in more detail Ostrom’s design principles and the relationship between managing common pool resources and g and ToM, which also follows the advice of Reviewer 1.

REVIEWER #3 REMARK:

2. There is a rich literature in psychology and organizational behavior that deals with the topic of group diversity; I point to some reviews that might help the authors (Bell, 2007; Mannix & Neale, 2005). In the current manuscript, there are several holes in the theoretical development and justification:

a. How are the authors thinking about diversity in this context? Is the highest level of functional diversity in a group achieved when there are both high levels of g and ToM, and hence both of these are resources that the group can use, or is the highest level of functional diversity achieved when there are high levels of one but low levels of another? A useful citation is Harrison & Klein, 2007 (Harrison & Klein, 2007).

REPLY:

We assume that the highest diversity in this case is reached when both, g and ToM are high. Those are also the cognitive skills needed to address the complex social-ecological issues and are hypothesized to increase the sustainable use of resources, especially when referred to what in economics are called rational and pro-social actors. In this revised version, we now make clear that functional diversity and ability level both matter. We have also worked to integrate the literature cited by the reviewer, where relevant, through the re-written Introduction and Discussion.

REVIEWER #3 REMARK:

b. Currently, the paper adopts a simplistic approach to aggregation to the group level, without consideration of the complex approaches and their theoretical justification. In small groups and teams research, much thought has been given to aggregating data to the group level. The following citations should be useful in justifying why the approach of taking the average was used (Barrick, Stewart, Neubert, & Mount, 1998; LePine, Buckman, Crawford, & Methot, 2011; Meslec, Aggarwal, & Curşeu, 2016).

REPLY:

We have added text to the main body of the manuscript in order to clarify why we chose to use average for general intelligence and minimum ToM as aggregation values for groups. The literature provided was relevant.

REVIEWER #3 REMARK:

c. There is work on collective intelligence or intelligence that emerges at the group level and cannot be attributed solely to the individual team members (and hence is different from the individual-aggregation level). While the work on collective intelligence has been cited, its implications have not been considered in the theoretical development of the paper. (Meslec et al., 2016; Woolley, Aggarwal, & Malone, 2015)

REPLY:

We have now added text that considers the emergence of a group level intelligence factor. The emergence of a group level intelligence that is predicted by ToM, but not g , is consistent with our theory development and results. ToM is a secret sauce that makes social groups more effective/efficient in their interactions; and this allows groups with higher ToM to perform better on a battery of tasks than groups with lower ToM. We have now added text specific to the emergence of group level intelligence in the Introduction/Theory and Discussion sections of the paper.

REVIEWER #3 REMARK:

3. Theoretical loose ends

a. While from the Abstract and the Discussion and Conclusion sections of the paper, the “long-term” versus “short-term” aspect of utilization of resources seems to be important, it is not mentioned nor developed in the front-end of the paper.

REPLY:

A common pool resource system is interesting precisely because individuals can either harvest now to maximize their short run benefit or control their harvest and share, which maximizes harvest over the longer run. This time component is thus built into the common pool resource system.

REVIEWER #3 REMARK:

b. Citations should be provided to qualify the claim that groups with higher g should be better at adapting to changes in their resource base as they are better at detecting changes in a system (Pages 3-4). Additionally, given the importance of change to the study (the experimental manipulation), there should be more about change and its association with group composition in the theoretical development.

REPLY:

Citations added.

REVIEWER #3 REMARK:

4. Contribution: As pointed out on Page 3, it seems that the FIP states that general intelligence and social intelligence are both capabilities that are critical for social groups to govern resources. Hence, is the contribution of this paper the interactive effect and not just the additive effect of the two types of intelligence? This should be presented clearly.

REPLY:

We now state this point directly. Please see Discussion section.

REVIEWER #3 REMARK:

Study

1. Was the sample size 54 groups? And 27 in each treatment? This information should be explicitly provided
2. The authors would need to defend that they had enough power to run six regression models per treatment with several control variables.

REPLY:

The sample size is 27 groups for the HL treatment and 25 groups for the LH treatment. This information is now included in the tables. The N for this study is not different from numerous experimental studies involving groups and related to common pool resource dilemmas (See Anderies et al. 2013, Baggio et al. 2015, Janssen et al. 2010 cited also in the main text for examples). Moreover, the sign and magnitude of the effects are consistent with the ones hypothesized via theory and literature. The major risk with respect to power here is, of course, Type 2 error. With respect to the major hypotheses, we obtained significant effects—thereby obviating the possibility of a Type 2 error. Observed power for the OLS regressions examining change in Tokens as a function of ToM, average g, and the interaction was .99 for the High to Low condition; the corresponding observed power for the Low to High condition was only .34—though was still statistically significant. We expected intelligence and ToM to matter more in the High to Low condition, and thus obtained high power where we expected it. Power was low for evaluating the control variables, but here, our major concern was not in obtaining effects, but rather ensuring that the predicted effects involving g and ToM did not result from unintended correlations with variables previously found to affect performance in resource depletion games. Finally, we would argue that we ran only 1 overarching model in each condition with predictor entered hierarchically so as to inform the reader as to whether or not the effects resulted from patterns of collinearity among those predictors. We still agree with the reviewer that the model is complex given that the small number of groups—but the major predictors represent emerge as statistically significant regardless of whether the model is simple (without control variables) or complex (with control variables).

REVIEWER #3 REMARK:

3. Currently, while I understand that the authors computed two dependent variables, the theoretical link between the two is unclear. Are they indicative of a similar theoretical construct? If not, could the authors build more up front in order to establish two distinct dependent variables.

REPLY:

Indeed they are indicative of the same theoretical construct, they both indicate the ability of groups to adapt to changing conditions. One assesses groups' ability to continuously harvest (and thus not deplete resources) the other variable assesses whether groups are increasing harvest pressure post vs pre changes in resources growth (or in the environment). However, although they both indicate the ability of groups to adapt to changing conditions, they the variables assess related but different aspects of performance. In general, the two metrics are more related in the High-to-Low outcome, where increases in ΔT leads to increases in time spent staring at an empty screen, or, in other words, an earlier depletion of resources. This is true overall, and when ΔT is measured against Time in each single round. That is, when conditions are pejorative and salience of g and ToM is greater (see figure 4 in the main text), then increasing harvest pressure (higher ΔT) is positively correlated with quicker resource depletion (Time). On the other hand, in the Low-to-High treatment, correlation between the two dependent variables is much weaker, except for round 4, and negative. That is, an increase in ΔT does not have an effect, or even has a positive effect on Time. These different results can be explained by the fact that in a Low-to-High setting, individuals, once they understand the dynamic of the system (often via learning by doing, we suspect), then they continue to harvest at that rate and do not adjust, or if they do, they increase harvest pressure while still being able to harvest sustainably.

Treatment	Round	Pearson	Spearman
Low-to-High	All	-0.0858	-0.1074
	1	0.117	0.1678
	2	-0.0412	-0.0923
	3	-0.0528	-0.1336
	4	-0.3551	-0.5421

	5	-0.276	-0.2154
	6	0.2566	0.0985
	All	0.2536	0.2363
	1	0.2853	0.189
	2	0.4984	0.3769
High-to-Low	3	0.3305	0.1492
	4	0.1296	0.1875
	5	0.2789	0.2033
	6	0.2566	0.2981

REVIEWER #3 REMARK:

4. From the Results section, the data showed a synergistic interactive effect in only the high-to-low condition, for the first dependent variable (Time). However, there is a compensatory interactive effect for both conditions with the second dependent variable (DD). These findings need to be reconciled and elaborated with much more cohesion.

REPLY:

The reviewer is correct, with the ΔT response variable, g and ToM have positive individual effects on ΔT and a negative interaction effect (compensating for the individual positive effects). Part of the issue here is that we were too loose with our use of the word synergy in the previous manuscript. Although the effect is one of “compensatory interaction,” it is important to recall the interpretation of the response variable. High values are “bad” (as they imply an increase in harvest pressure), and low values are “good” (as they imply a decrease or constant harvest pressure). What we find is that when you change the resource system, if a group has low g or low ToM, they push the ecological system much closer to its boundary risking collapse. If both g and ToM are high, groups seem to recognize the change and harvest in a much more sustainable way. Colloquially, g and ToM are in synergy in this instance as both improve the sustainable use of the resource. Technically, the reviewer is correct, this is not a synergy at all but a compensatory effect for the bad outcomes related to only having high g or only having high ToM. In this revised version of the manuscript, we have taken care to use more precise language in our description of the results and in the discussion. This result is perfectly consistent with the FIP in that both traits are needed in conjunction to adapt to perturbations to a common pool resource system.

REVIEWER #3 REMARK:

5. Suggestion: When computing the second dependent variable, instead of using the average of the three times before and after the treatment and then calculating the difference, could the authors calculate the slope using OLS regression for the three times before and the three times after and calculate the difference in the slopes? Would that measure capture the richness of the longitudinal data in a richer way? Does it yield similar results? The first round’s scores will still need to be controlled for.

REPLY:

While using the slope could capture changes in importance of ToM and g in each round, this result would not differ from the Time dependent variable, although in that case, a logistic form is used. With ΔT we capture the actual average difference in pressure pre-post perturbation, hence round-to-round variations are attenuated giving us a better understanding of the relationship between cognitive abilities and group ability while adapting to changes in harvests. We average the three rounds to account for communication importance in solving the pressure and depletion issues, as communication is related to ToM.

REVIEWER #3 REMARK:

Discussion and Conclusions

1. The Discussion section needs to be much more crisp and attentive to the results; for example on Page 14 (and 18), the authors write “we reason..... and our analysis is consistent with FIP.” The authors only found a synergistic interactive effect for one of the treatment conditions and only for one of the dependent variables (Time). The authors need to be very precise in the deductions they make from the results to avoid overclaiming.

REPLY:

In this revised version of the manuscript, we are now more precise in our description of the results and where the results are and are not consistent with theory following the advice of the reviewer.

REVIEWER #3 REMARK:

2. As pointed above, much more can be done to tie together the two dependent variables and establish what theoretical variable they are capturing; further the short-term, long-term distinction, while appealing, is not currently developed adequately, and several concerns are raised in the reader’s mind. For example, can a short-long term claim really be made given the very short time scale of the experiment (where each round was 180 seconds).

REPLY:

Indeed we agree and have now rewritten the discussion and results section (as well as most of the manuscript) following the reviewers advice. WE do agree that the short-long term distinction is artificial and have avoided such over-claims. We also have sharpened the discussion on why and how the two dependent variables are linked. The two dependent variables chosen, while capturing different aspect of potential issues in harvesting resources sustainably, are theoretically related. In fact, while time measures the speed of collapse (and whether group actually completely deplete resources and how much time they employ to deplete resource, if they do), ΔT measures how group harvest pre-vs-post perturbation. Increasing harvest pressure in degraded environments is detrimental to the longer-term sustainability of resources, that is, increased harvest pressure relates to an increase in the probability of a group collapsing resources (see also the correlation table above, where the two dependent variables are correlated especially when the environment has pejorative changes). This is however, not evident in the low-to-high treatments, that is, when the environment allows for more resource harvest. In this latter case, an increase in resource pressure has no consequences on resource depletion, and correctly so, given that the environment allows for increased harvest pressure, as resources are regenerating faster.

REVIEWER #3 REMARK:

3. External validity: the authors need to convince the audience of the external validity of this research question.

REPLY:

(The following paragraph has been also added in the SI)

All experimental studies have issues with external validity, more so if performed in a western university with an undergraduate population. We compare here our average results with the wider U.S. population using data available from the Gallup Report for 2016 available here: <https://news.gallup.com/poll/200186/five-key-findings-religion.aspx>. We find that our religious diversity is very much in line with the average religious diversity within the wider U.S. population. In fact, our average religious diversity metric 0.82 (see table 1 in the supplementary material) is very similar to the average religious diversity metrics of the U.S. when all Christian religions are considered together (0.84), albeit lower when different christian religions are separated, as diversity then increases to 1.38 for the U.S. population. Comparing the ethnic diversity of our sample size with the wider U.S. population we find that our sample composition is more homogeneous than the overall U.S. population, however, this also reflects the inbuilt biases that exist within the U.S. college population. In fact, while our ethnic diversity index (calculated as described in the main paper, materials and method section) is = 0.53 on average, the U.S. population ethnic diversity index (dividing white and Hispanics) is = 1.1 – almost double. However, if we look at the number of groups of specific diversity, while most groups are very homogeneous, over 30% of our groups reflect the overall U.S. average

Ethnic Diversity Index	Percent
---------

0	36.54
0.5623351	25.00
0.6931472	7.69
1.039721	28.85
1.386294	1.92

With respect to age, our sample is definitely skewed compare the wider population given the age restriction and inbuilt bias that exist when sampling undergraduate students.

Unfortunately, no data are really available for ToM. In a previous study Freeman and colleagues (2016) have used agreeableness as a proxy to ToM, however, the comparability between the metric we use in this study, based on a specific reasoning test, and the agreeableness metric used may not be comparable. On the other hand, our sample, as expected, has a narrower spreading as well as higher g compared to population average with respect to the U.S.

Overall, while not perfect, our sample fairly accurately reflects the wider population of interest.

REVIEWER #3 REMARK:

Suggested references:

Barrick, M. R., Stewart, G. L., Neubert, M. J., & Mount, M. K. (1998). Relating member ability and personality to work-team processes and team effectiveness. *Journal of Applied Psychology*, 83(3), 377.

Bell, S. T. (2007). Deep-level composition variables as predictors of team performance: a meta-analysis. *Journal of Applied Psychology*, 92(3), 595.

Harrison, D. A., & Klein, K. J. (2007). What's the difference? Diversity constructs as separation, variety, or disparity in organizations. *Academy of Management Review*, 32(4), 1199–1228.

LePine, J. A., Buckman, B. R., Crawford, E. R., & Methot, J. R. (2011). A review of research on personality in teams: Accounting for pathways spanning levels of theory and analysis. *Human Resource Management Review*, 21(4), 311–330.

Mannix, E., & Neale, M. A. (2005). What differences make a difference? The promise and reality of diverse teams in organizations. *Psychological Science in the Public Interest*, 6(2), 31–55.

Meslec, M. N., Aggarwal, I., & Curşeu, P. L. (2016). The insensitive ruins it all: Compositional and compilational influences of social sensitivity on collective intelligence in groups. *Frontiers in Psychology*, 7, 676.

Woolley, A. W., Aggarwal, I., & Malone, T. W. (2015). Collective Intelligence and Group Performance. *Current Directions in Psychological Science*, 24(6), 420–424.

Wynne, C., D. L. (2001). *Animal Cognition: The Mental Lives of Animals*. Palgrave Publishers.

Reviewer #4 (Remarks to the Author):

This is a very interesting paper. The research question is strong and interesting, and it is timely given this ever important topic (sustainability). Functional diversity is an important ingredient in maintaining sustainability. The authors have done a very good job of isolating and measuring the two variables of interest (general intelligence and social intelligence). I thought that using the SAT/ACT scores from participants to estimate general intelligence was particularly well done. Furthermore, their statistical analysis is good and the robustness checks strengthen their results, though I have a few questions and suggestions below.

REPLY:

We thank the reviewer for the points raised and we are convinced that the reviewer will find this new version of the paper much improved.

REVIEWER #4 REMARK:**MAJOR POINTS:**

1) I looked at your models carefully and liked the robustness checks that you conducted: controlling for chat, trust, religious and ethnic diversity, and gender are all useful in providing good evidence of your phenomenon. However, one IV I couldn't find (but seems rather important) is an indicator variable for the change of growth rate within each treatment. That is, I'd expect a "\delta growth" indicator variable and a (three-way) interaction with ToM and g in your models. After all, if I understand your argument correctly, you are proposing that ecological change (i.e. the missing indicator variable) has differential effects depending on the ToM and g of your groups. Correct?

REPLY:

Indeed, however, the change is always the same for all groups in both experiments. In the high-to-low, we reduce the growth by half, and in the low-to-high, we increase the growth doubling it. That is why we run two separate analyses. In order to robustly address the comment, however, please see below the results with a three-way interaction. The actual growth rate seems to matter less, while the compound effect of g and ToM is still significant and important in reducing the time participants spend staring at depleted resources. In other words, g*ToM increases resource sustainability. See below the first table represent three way effects on time-left, and the second table represents three way effects on ΔT .

AIC	222.828	224.614	226.580	228.526	228.826	230.806
Dispersion	0.295	0.295	0.296	0.297	0.292	0.293

Dep var = ΔT Indep Var:	grr1	grr2	grr3	grr4	grr5	grr6
ToM	0.065*** (0.015)	0.065*** (0.015)	0.069*** (0.015)	0.066*** (0.017)	0.066*** (0.017)	0.051*** (0.018)
g	0.023*** (0.005)	0.023*** (0.005)	0.025*** (0.005)	0.022*** (0.006)	0.022*** (0.006)	0.017** (0.007)
ToM*g	-0.004*** (0.001)	-0.004*** (0.001)	-0.004*** (0.001)	-0.004*** (0.001)	-0.004*** (0.001)	-0.003*** (0.001)
LH	-0.646*** (0.190)	-0.628*** (0.189)	-0.626*** (0.188)	-0.704*** (0.193)	-0.704*** (0.193)	-0.893*** (0.242)
LH*ToM	0.009 (0.029)	0.007 (0.029)	0.006 (0.030)	0.016 (0.031)	0.016 (0.031)	0.044 (0.037)
LH*g	0.021** (0.009)	0.020** (0.009)	0.020** (0.008)	0.024*** (0.009)	0.024*** (0.009)	0.032*** (0.011)
LH*TM*g	-0.000 (0.001)	-0.000 (0.001)	-0.000 (0.001)	-0.001 (0.001)	-0.001 (0.001)	-0.002 (0.002)
D Chat		-0.001 (0.002)	-0.001 (0.002)	0.000 (0.002)	0.000 (0.002)	-0.001 (0.003)
Trust			-0.002 (0.003)	-0.002 (0.003)	-0.002 (0.003)	-0.003 (0.004)
Religious Diversity				-0.030* (0.017)	-0.030* (0.017)	-0.017 (0.031)
Ethnic Divdersity						-0.040* (0.020)
% Male						0.075 (0.059)
Constant	-0.238** (0.109)	-0.238** (0.109)	-0.272** (0.122)	-0.194 (0.143)	-0.194 (0.143)	-0.103 (0.145)
N	312.000	312.000	312.000	312.000	312.000	312.000
AIC	-521.217	-519.319	-517.518	-519.643	-519.643	-526.640
R^2	0.447	0.447	0.447	0.454	0.454	0.473

REVIEWER #4 REMARK:

In short, I think the following ought to be clarified: what is the effect of the growth rate change (as described above) and, equally important, what would be your predictions for such a three-way interaction? I'd spend some more time/space in the introduction clarifying what you expect to see in your different treatments and why.

REPLY:

As shown in the tables above, cognitive abilities have a significant effect, and most of all, the compound effect (ToM * g) is what really is important for the sustainability of resources. This result is consistent with the analysis done throughout this work, where we separated the two treatments (with low to high and high to low growth rates). For obvious reasons, the dummy variable related to growth rate is also significant when it comes to assessing the difference between pre-post perturbation average harvest.

REVIEWER #4 REMARK:

2) My second main question is around the choice of metrics in our main models. You chose to focus on the smallest social intelligence in each group ($\min(\text{ToM})$) and the average general intelligence in each group ($\text{avg}(g)$) in your main model (Tables 2 and 3 in the SI). It seems like an arbitrary choice to use "min" for one metric and "avg" for the other, and it was not motivated in the introduction/theory section. To be honest, I'm not familiar with any literature that argues convincingly that either min or avg are the appropriate measures of general/social intelligence in groups to use, but perhaps there is – it seems important to look into this.

REPLY:

Indeed the reviewer is right, we have added a clear justification on why we use average g and min ToM as this was also brought up, correctly so, by another reviewer as well. We now provide a theoretical solid justification of why, in the main analysis, we include avg g and min ToM as group level metrics of individual g and ToM. Such justification is rooted in the different tasks that g and ToM aid individuals and groups to perform. g can be thought of an aggregate task: understanding the system, I understand a bit + you understand a bit more + someone understand a lot, hence our total understanding is = to an average, in our case, as all groups have the same number of people). ToM on the other hand relates to mental models and communication as well as diffusing conflict; in other words, ToM relates to understanding and anticipating social uncertainty. In this latter case, the task that ToM helps with can not be solved by aggregation, but by conjunction, that is, is more akin to a chain, where everything works until the weakest link gives up. In fact, it often takes only the weakest link to increase conflict and cost of communication tremendously (see also the work cited in the main text and suggested by Reviewer 3 by Barrick et al in 1998 and Meslec et al. in 2016).

REVIEWER #4 REMARK:

On a related note, I appreciate that the authors included other variants of the model specification, e.g. $\text{avg}(g) * \text{avg}(\text{ToM})$ and $\min(g) * \text{avg}(\text{ToM})$ and $\min(g) * \min(\text{ToM})$ in the SI (Tables 4-7). And while not all models show a significant interaction, the evidence in aggregated across the tables seems to be consistent with the predicted significant interaction, so that makes sense. That said, I'd push the authors to think harder about which metrics (avg or min) should be used and why, and use that as the main table to refer to in the main text. Unless there is a good reason for it, I'd suggest sticking with either min for both or avg for both.

As a purely empirical, post-hoc observation to stimulate your thinking, I found it intriguing that the models using $\min(g)$ and $\min(\text{ToM})$ were consistently and significantly in the predicted direction – one way to read this (again, caution because post-hoc interpretation) is that the "weakest link" in the group matters: the lowest general and social intelligence is a good predictor of how well the group does.

REPLY:

Indeed this is correct, and we also reference to this in the main text: ". One individual with very low \$ToM\$ has the ability to jeopardize the effectiveness of the whole group (Meslec et al. 2016)".

REVIEWER #4 REMARK:

3) Lastly, I understand your arguments around the synergetic effects of ToM * g. However, I couldn't quite understand the main effects of those variables. That is, at the moment there are no regression models with just ToM and g as main effects (without the interaction). That would be a first important addition.

REPLY:

Indeed the reviewer is correct, however, we also would like to note here that our hypothesis is that there is a compounding effect of the functional diversity of cognitive abilities given by both g and ToM. Without the interaction effect, this is not possible to assess and the main hypothesis of this work would not be tested. Hence, we added the following tables in the supplementary material only. Without a compound effect, both g and ToM reduce the time stared at the screen. That is, the higher both are, the more a group harvests sustainably. ToM, however, has 3 times the effect of g for time and 5 for pressure on resources before and after the perturbation (ΔT). However, in the Low to High treatment, when resources actually positively change, ToM reduces pressure, but g is the main driver for avoiding resource depletion, albeit often not significantly.

Dependent variable: Time, treatment : High to Low.

	grt1	grt2	grt3	grt4	grt5	grt6
ToM.	-0.301*** (0.062)	-0.301*** (0.063)	-0.318*** (0.069)	-0.318*** (0.069)	-0.305*** (0.062)	-0.312*** (0.063)
G.	-0.101* (0.052)	-0.101** (0.051)	-0.131** (0.051)	-0.138*** (0.052)	-0.128** (0.054)	-0.125** (0.054)
1.round	0.000 (.)	0.000 (.)	0.000 (.)	0.000 (.)	0.000 (.)	0.000 (.)
2.round	-1.369*** (0.440)	-1.370*** (0.435)	-1.364*** (0.434)	-1.366*** (0.435)	-1.413*** (0.415)	-1.420*** (0.416)
3.round	-1.706*** (0.453)	-1.707*** (0.465)	-1.697*** (0.463)	-1.700*** (0.461)	-1.764*** (0.431)	-1.772*** (0.429)
4.round	-1.546*** (0.471)	-1.547*** (0.453)	-1.537*** (0.448)	-1.540*** (0.451)	-1.601*** (0.437)	-1.611*** (0.428)
5.round	-1.885*** (0.395)	-1.886*** (0.402)	-1.875*** (0.399)	-1.879*** (0.401)	-1.949*** (0.384)	-1.959*** (0.382)
6.round	-2.276*** (0.453)	-2.277*** (0.479)	-2.266*** (0.470)	-2.270*** (0.473)	-2.350*** (0.484)	-2.359*** (0.485)
D chat		0.000 (0.043)	-0.003 (0.041)	-0.002 (0.042)	0.005 (0.039)	0.007 (0.038)
trust			0.100 (0.087)	0.102 (0.086)	0.184** (0.090)	0.178** (0.087)
Rel Div				-0.166 (0.313)	-0.868** (0.404)	-0.996* (0.508)
Ethn Div					1.226** (0.479)	1.302** (0.545)
%male						0.324 (0.719)
cons	3.702*** (1.140)	3.700*** (1.164)	4.547*** (1.135)	4.857*** (1.247)	4.470*** (1.354)	4.302*** (1.390)
N	162.000	162.000	162.000	162.000	162.000	162.000
aic	103.783	105.783	107.366	109.295	109.279	111.223
dispers	0.224	0.225	0.224	0.225	0.213	0.214

Dependent Variable: ΔT , Treatment: High to Low

	grr1	grr2	grr3	grr4	grr5	grr6
ToM.	-0.015*** (0.002)	-0.015*** (0.002)	-0.013*** (0.002)	-0.013*** (0.002)	-0.013*** (0.003)	-0.013*** (0.003)
g	-0.003** (0.001)	-0.003** (0.001)	-0.001 (0.002)	0.000 (0.002)	-0.000 (0.002)	-0.000 (0.002)
D Chzat		0.000 (0.002)	0.003* (0.002)	0.002 (0.002)	0.003 (0.002)	0.003** (0.002)
Trust			-0.010*** (0.002)	-0.009*** (0.002)	-0.009*** (0.002)	-0.009*** (0.002)
Religious Div				0.014 (0.010)		0.025 (0.015)
Ethnic Div					0.010 (0.011)	-0.008 (0.016)
% male						-0.030 (0.026)
cons	0.323*** (0.028)	0.322*** (0.027)	0.249*** (0.034)	0.227*** (0.045)	0.239*** (0.038)	0.234*** (0.046)
N	162.000	162.000	162.000	162.000	162.000	162.000
aic	-484.209	-482.228	-494.811	-494.916	-493.553	-492.491
r2	0.339	0.339	0.396	0.404	0.399	0.410

Dependent variable: Time, treatment : Low to High.

	grt1	grt2	grt3	grt4	grt5	grt6
--	------	------	------	------	------	------

ToM.	0.046 (0.063)	0.044 (0.063)	0.044 (0.061)	0.016 (0.066)	0.027 (0.066)	0.027 (0.066)
G	-0.134** (0.061)	-0.134** (0.060)	-0.135** (0.067)	-0.103 (0.075)	-0.092 (0.078)	-0.094 (0.076)
1.round	0.000 (.)	0.000 (.)	0.000 (.)	0.000 (.)	0.000 (.)	0.000 (.)
2.round	-1.518*** (0.460)	-1.337*** (0.494)	-1.337*** (0.494)	-1.291*** (0.499)	-1.286*** (0.496)	-1.303*** (0.495)
3.round	-1.229*** (0.413)	-1.063** (0.440)	-1.063** (0.440)	-1.020** (0.445)	-1.017** (0.451)	-1.033** (0.459)
4.round	-1.963*** (0.441)	-1.860*** (0.453)	-1.860*** (0.453)	-1.837*** (0.458)	-1.839*** (0.453)	-1.850*** (0.458)
5.round	-2.292*** (0.449)	-2.182*** (0.462)	-2.183*** (0.462)	-2.158*** (0.471)	-2.156*** (0.470)	-2.168*** (0.475)
6.round	-2.414*** (0.496)	-2.353*** (0.498)	-2.353*** (0.498)	-2.343*** (0.507)	-2.342*** (0.508)	-2.348*** (0.515)
D chat		-0.044 (0.039)	-0.044 (0.039)	-0.057 (0.040)	-0.059 (0.039)	-0.055 (0.045)
Trust			0.003 (0.082)	-0.012 (0.083)	0.000 (0.085)	0.009 (0.095)
Rel Div				0.511 (0.389)	0.337 (0.507)	0.328 (0.516)
Ethn Div					0.308 (0.444)	0.294 (0.439)
%male						0.199 (0.942)
cons	2.627** (1.325)	2.797** (1.314)	2.818* (1.516)	1.915 (1.748)	1.597 (1.800)	1.529 (1.894)
N	150.000	150.000	150.000	150.000	150.000	150.000
aic	125.762	127.255	129.254	130.623	132.424	134.403
dispers	0.383	0.382	0.385	0.383	0.384	0.387

Dependent Variable: ΔT , treatment: Low to High

	grr1 b/se	grr2 b/se	grr3 b/se	grr4 b/se	grr5 b/se	grr6 b/se
ToM.	-0.012** (0.005)	-0.010** (0.005)	-0.008* (0.004)	-0.002 (0.003)	-0.008** (0.003)	-0.007* (0.004)
g	0.016*** (0.003)	0.015*** (0.003)	0.011** (0.004)	0.004 (0.006)	0.001 (0.005)	-0.002 (0.005)
D chat		-0.007 (0.006)	-0.006 (0.006)	-0.004 (0.004)	-0.006 (0.004)	-0.006 (0.004)
Trust			0.013*** (0.005)	0.017*** (0.005)	0.011** (0.004)	0.022*** (0.008)
Rel Div				-0.113*** (0.041)		-0.052 (0.045)
Ethn Div					-0.123*** (0.023)	-0.112*** (0.034)
%male						0.229** (0.096)
cons	-0.275*** (0.074)	-0.278*** (0.074)	-0.173* (0.101)	0.044 (0.150)	0.110 (0.114)	0.063 (0.104)
N	150.000	150.000	150.000	150.000	150.000	150.000
aic	-158.021	-157.216	-158.717	-170.037	-174.739	-187.822
r2	0.101	0.108	0.129	0.203	0.228	0.311

REVIEWER #4 REMARK:

Second, I couldn't quite make sense of the single effects of ToM and g in your existing models. E.g. looking at Table 3, it makes sense that ToM*g is significant and negative (i.e. reducing the amount overharvested). But why is the single effect of ToM (0.065***) and g (0.023***) positive? I think it may make sense to plot the actual data on a ToM vs. g heatmap grid, and see what the effects in each corner are like when independently varying just one variable, and explaining this to the reader (perhaps the marginal plots are already doing this, but I couldn't quite follow it – I felt the labels and titles in the graphs were not very easy to understand on their own).

REPLY:

Indeed the figures already indicate marginal effects. Basically what is happening is that increased levels of g and/or ToM by themselves negatively affect the resource sustainability, or the ability of groups to harvest sustainably. However, combined, they increase the ability of groups to adapt to changing conditions as well as the ability of groups to harvest sustainably. In other words, our results highlight the importance of functional cognitive diversity and the functional intelligence proposition.

REVIEWER #4 REMARK:**MINOR POINTS:**

* I liked your Figure 3 but felt that it came way too late in the text. I'd move this upwards into the introduction to give people a better sense of the magnitude of the problem you are trying to tackle. By the way, incidentally, I would encourage you to use the same figure to add in the number of groups (N=...) that you have that are low g/low ToM, low g/high ToM, high g/low ToM and high g/high ToM, so that readers can immediately see that you cover the whole spectrum of interest in your experiment/analysis.

REPLY:

We thank the reviewer for the suggestion, we have heavily thought about moving figure 3 earlier, however, given the amendments to the text and the insertion of figure 1 we think that the paper has drastically improved and that Figure 3 (now figure 4) is in the correct position in the paper.

REVIEWER #4 REMARK:

* I thought your argument to include round number in the analysis was sensible. Why did you only include it in the models predicting time spent thinking, but not in the ΔT models? I'd add them in the latter too.

REPLY:

Unfortunately we are not able to add round number for the ΔT . Models, as the dependent variable is the average token harvested pre vs post perturbation. As such, the analysis here is performed via OLS and rounds are not possible to assess (they are collapsed into the ΔT dependent variable).

REVIEWER #4 REMARK:

* Personally, I think that the evidence you present on ΔT was more convincing than the time spent thinking about the resource. This is because some literature (e.g. Evans, Dillon & Rand, 2015) would suggest that the latter (decision time) does not indicate the mental state of individuals (not sure there's evidence of teams) but rather indicates the extent to which an individual faces decision conflict. So to the extent that you want to demonstrate that teams exercise restraint in their self-interested choices, I'd suggest that ΔT is a better proxy than decision time.

REPLY:

We would like to clarify that Time = time spent staring at an empty screen, in other words, Time = how fast resources are depleted. When Time = 1 resources are depleted immediately (the whole group overharvests super fast), when time = 0 resources are not depleted and the group is able to harvest until the end of a round. We are sorry for causing the misunderstanding, we hope to have clarified the meaning of the "time" variable.

REVIEWER #4 REMARK:

* Clarification question of how you define "time spent looking at the screen" (one of your outcome variables): given that in low-growth environments a resource takes longer to re-grow, I assume you normalize the "time spent" variable by the time it takes a resource to grow to full capacity, correct? (Otherwise, it seems you'd bias the low-to-high treatment against the other due to slow start/learning effects in earlier rounds.)

REPLY:

Please see comment above. Time = time actually that occurs for resources to be depleted. The variable is rescaled so that at value 0 = resources are not depleted and groups can harvest until the end of the round. 1 = resources are depleted immediately, groups will spend most of the round not being able to harvest tokens and staring at an empty screen.

REVIEWER #4 REMARK:

* At the moment you do not compare the two treatments (HL vs. LH) in your models (i.e. using treatment assignment as an IV), so I'd be careful with the language you use to describe the results. E.g. "Figure 1 illustrates that groups with both high g and high ToM are less likely to collapse their resource, especially in the high-to-low treatment, before a round is finished."  The result you refer to is actually only true in the high-to-low treatment, not "especially" – this may be a minor point but I'd be precise in your language based given your models.

REPLY:

Indeed we corrected the text to reflect our results more rigorously throughout the text.

REVIEWER #4 REMARK:

* How were people treated in the analysis who were familiar with the SST? If those familiar are excluded from the analysis, what happens if they are included?

REPLY:

To the best of our knowledge, None of the individuals participating in the experiments was familiar with SST. In fact, we have an oblique way to measure this. Part of the SST protocol is to ask participants if they have ever read the Hemingway story that the test is based on. Individuals could not be exposed to the test without being exposed to the story. Less than 2 percent of participants reported previous exposure to the story.

REVIEWER #4 REMARK:

* A recently published paper in Nature might be of relevance and should probably be included in the discussion of cognitive (intellectual) and social drivers of cooperation: González-Forero, M., & Gardner, A. (2018). Inference of ecological and social drivers of human brain-size evolution. *Nature*, 557(7706), 554.

REPLY:

We thank the reviewer for the suggestion and indeed the paper was instrumental to strengthen our arguments in relation to group intelligence, and g and ToM.

Reviewers' comments:

Reviewer #1 (Remarks to the Author):

The authors have adequately addressed all of my comments. I believe the paper is in publishable-shape from my perspective.

Reviewer #3 (Remarks to the Author):

It is evident that the authors have re-written the manuscript with careful analysis and a more comprehensive review of the associated literatures. The authors have addressed the concerns raised by Reviewer 3 in a thoughtful and deliberate way, both in the response letter and the manuscript. In the manuscript, this is visible, for example, in the addition of more precise definitions, differentiation of ToM and g, work on development of the rationale behind the interaction (including Figure 1), and elaborating on and clarifying the previously underdeveloped theory. The citations provided were also incorporated in the intended manner. Additionally, the Discussion and Conclusion sections have been rewritten, with a clearer and more precise interpretation of the results. It is clear that the authors have put in directed effort, and worked with the comments in an engaged and thorough manner. The resulting manuscript is interesting, precise, and theoretically well developed.

Reviewer #4 (Remarks to the Author):

Review of revision of FIP and sustainability paper

This paper has improved in several respects (e.g. addition of Figure 1, clarifications of certain time construct, addition of theoretical motivation for min/avg metrics). However, while it has been improved, I think the authors did engage too much in "hand waving" to address some of the concerns raised by me (and the other reviewers, e.g. Ref. 3 #3b). I may have been too suggestive and not instructive enough in my initial review, so I will now make clearer what I think are important and necessary changes to back up the claims made, without which I can't recommend publication since your claims are not yet fully backed up by the evidence. I would thus ask that the authors more clearly address the following points and make the requested changes in the manuscript:

Most importantly, the authors make a strong claim in the abstract and introduction that the environment fluctuations matter for their predictions – i.e. that the high vs. low and low vs. high environmental conditions have an effect on harvest and resource collapse. For example, P. 5: "In the case of continuous and sudden changes within the environment, the interaction effects of g and T oM should be even more pronounced." In fact, you dedicate a full paragraph between pages 5-6 to explaining how the environment change (shock) will have effects on FIP and group outcomes, but you lack two important parts to make this claim (and I've added two additional clarifications at the end):

(1) The first problem is that there are no clear predictions: i.e. what are the **specific** predictions going from low-to-high and high-to-low, and how are those low-to-high predictions **different** going from high-to-low? I doubt that a resource change acts the same way whether it goes from "all is good" to "all is bad" or vice versa. There are likely very different. In fact, I also echo one of the other reviewers' call that you need to provide more of a theoretical grounding, including for those environmental shocks: what does the past literature and existing theory tell us about which way these effects are predicted to go? How would a low-to-high shock look different from a high-to-low shock?

* By the way, you need to introduce the low-high and high-low experimental design earlier in the paper: at the moment, I can't tell in which round the resource change happens in either treatment – is it round 3 or 4, or a different round? (It didn't find this information either in "Materials and Methods" or the SI.) Moreover, you mention the treatments before you define them: on page 9, last paragraph, you start a sentence with "In both the high-to-low and low-to-high treatments, ..." without having explained or defined the two treatments earlier.

* Let me clarify something else (or make sure that this is in fact what the authors were intending to get across): in the previous manuscript it sounded like you were contrasting the low-to-high and high-to-low treatments as an empirical exercise, so the interaction in the response letter you provided was the answer to this. But the revision of your main text (and my re-reading of it) suggests that you are not so interested in comparing low-to-high with high-to-low side by side – which is fine, and the revised text doesn't suggest that anymore. HOWEVER, this revision has made clear that you really need to show something about the two different treatments and their (presumably differential) effects on your outcomes. So, instead of doing a between-treatment comparison, you should focus on a "shock to the system" within each treatment and make clear what you expect the effects of low-to-high and high-to-low to be and then test those (see next point).

(2) The second problem is that these low-to-high and high-to-low predictions are not tested explicitly with the regressions. Specifically, you need to be clear what you are saying in the text matches what actually test with regressions. For example, this phrase in the abstract "... even after a shock to the resource system." lets the reader assume that you will test the effect of FIP after (i.e. dummy variable shock=1) a change in environment. This should be tested explicitly in a regression model; at the moment I can't find this anywhere.

* To be clear, from your discussion and specifications of your other models, this does require a three way interaction of the following format: $g * ToM * shock$ for both the low-to-high and high-to-low treatments separately. And three-way interactions are not easy to understand or interpret, so care needs to be taken. In fact, I would recommend you have a table with at least 4 columns (no need for control variables in each of them, maybe add one additional column if you want that includes all control variables at once, but there's no need to have every table have each controls variable added incrementally). So, I ask that you include the following table, first for low-to-high:

Column 1: your standard regressions $g * ToM$ plus a control variable for shock=1 (no interaction with $g * ToM$ yet).

Column 2: same regression ($g * ToM$) but this time only the subsample for which shock=0 (i.e. show us the effect of $g*ToM$ before the shock occurred)

Column 3: same regression ($g * ToM$) but this time only the other subsample where shock=1 (i.e. after the shock occurred)

Column 4: full three-way interaction model $g * ToM * shock$. This is a big table and will need careful explaining, so don't hand-waive this: instead think carefully what columns 2 and 3 tell you separately and then try to understand and explain column 4 – a three-way interaction is not easy. (Optional column 5: same as Column 4 but with all the control variables at once if you like.)

* And of course you will need another table like this for high-to-low. Again, I would be surprised if the effects are identical for those two tables since going from low-to-high and high-to-low probably shouldn't be identical, as the experience is completely different. You will need to develop a good theoretical grounding and argument for this (see point 1 above).

(3) On another note, I have noticed that you have made some improvements to explain what is going on in your experiment and what the results are. But I think you will need to do more to

explain your figures and your findings. The other reviewers and I had both questions about the marginal effects and what g and ToM independently do: it was great that you added Figure 1 to explain this more easily, but please also spend more time to explain what the results mean. At the moment, your paper feels more like "trust that the authors are interpreting the results correctly" than explicitly telling the reader what the coefficients mean and how you arrive at your conclusion – and this is a problem, because as a reader, we can't verify this sort of statement:

"This allows us to track, on average, whether groups push the ecosystem harder (a high positive value), ease their harvest (a negative value), or remain the same (a value of zero) after an ecological change."

Where do you read that from? Please give more evidence and use the regressions to explicitly test your hypothesis before drawing conclusions that are hard to verify (and may or may not be true, I don't know yet).

(4) I recommend that you take advantage of the Supplementary Information more: at the moment you have some regression tables there, a bit of methods detail, and the instructions but I think it would pay off to use the space more – you have lots of space in the SI, so please do much of the explaining I asked for above in the SI (with all the details, coefficient interpretations and p-values!) and then take what is important and transfer it into the main text (with clear cross-references to the corresponding tables in the SI and with clear interpretation of the regressions you are testing there).

In sum, this remains an interesting and timely paper but I think the authors need to do more work to (1) clarify their hypotheses, (2) make sure their text matches what they test statistically, (3) use less hand-waiving to articulate their findings and (4) add more detail in the SI to convince readers that they have the evidence they claim they have. I wish the authors good luck for the next revision and I look forward to the next version of the manuscript.

Reviewers' comments:

Reviewer #1 (Remarks to the Author):

The authors have adequately addressed all of my comments. I believe the paper is in publishable-shape from my perspective.

REPLY

We would like to thank the reviewer for the stimulating remarks. Thanks to the reviewer's comments we were able to clarify the important linkages existing between cognitive abilities and group ability to manage resources sustainably as well as strengthen the links between cognitive abilities and the common pool resource literature. We think that this line of research is promising and can improve how we approach the governability of resources for the challenges that lie ahead.

Reviewer #3 (Remarks to the Author):

It is evident that the authors have re-written the manuscript with careful analysis and a more comprehensive review of the associated literatures. The authors have addressed the concerns raised by Reviewer 3 in a thoughtful and deliberate way, both in the response letter and the manuscript. In the manuscript, this is visible, for example, in the addition of more precise definitions, differentiation of ToM and g, work on development of the rationale behind the interaction (including Figure 1), and elaborating on and clarifying the previously underdeveloped theory. The citations provided were also incorporated in the intended manner. Additionally, the Discussion and Conclusion sections have been rewritten, with a clearer and more precise interpretation of the results. It is clear that the authors have put in directed effort, and worked with the comments in an engaged and thorough manner. The resulting manuscript is interesting, precise, and theoretically well developed.

REPLY

We would like to thank the reviewer for stimulating us to really tighten and strengthen our paper. The suggestions and comments made really helped clarify the theoretical linkages between cognitive abilities and managing common pool resources sustainably. We do think that this line of research can be further explored and will be important for the challenges that lie ahead in our society.

Reviewer #4 (Remarks to the Author):

Review of revision of FIP and sustainability paper

This paper has improved in several respects (e.g. addition of Figure 1, clarifications of certain time construct, addition of theoretical motivation for min/avg metrics). However, while it has been improved, I think the authors did engage too much in "hand waving" to address some of the concerns raised by me (and the other reviewers, e.g. Ref. 3 #3b). I may have been too suggestive and not instructive enough in my initial review, so I will now make clearer what I think are important and necessary changes to back up the claims made, without which I can't recommend publication since your claims are not yet fully backed up by the evidence. I would thus ask that the authors more clearly address the following points and make the requested changes in the manuscript:

REPLY

We appreciate the reviewer's constructive comments, which have strengthened the paper. We have addressed all suggestions and modified the text accordingly. The changes (detailed below) clarify our argument and the theoretical motivation for the study-

Most importantly, the authors make a strong claim in the abstract and introduction that the environment fluctuations matter for their predictions – i.e. that the high vs. low and low vs. high environmental conditions have an effect on harvest and resource collapse. For example, P. 5: "In the case of continuous and sudden changes within the environment, the interaction effects of g and ToM should be even more pronounced." In fact, you dedicate a full paragraph between pages 5-6 to explaining how the environment change (shock) will have effects on FIP and group outcomes, but you lack two important parts to make this claim (and I've added two additional clarifications at the end):

REPLY

We do make such a claim, and we have tried to clarify why (see below). Further, our experiment spans different domains from simple to complex in both the ecological and social domain. In fact, the low-to-high treatment can be thought of a simpler problem and, as such, the importance of high competence in both g and ToM is less pronounced (if any importance at all). Such problems are simple to solve, as one can rely on past experiences and avoid resource collapse. However, in the high-to-low treatment, changes are negative and such changes require re-negotiation and understanding the new dynamics in order to adjust group and individual harvest strategies. In this situation, a functional diversity of cognitive abilities becomes key in order to maintain a sustainable harvest.

(1) The first problem is that there are no clear predictions: i.e. what are the *specific* predictions going from low-to-high and high-to-low, and how are those low-to-high predictions *different* going from high-to-low? I doubt that a resource change acts the same way whether it goes from "all is good" to "all is bad" or vice versa. There are likely very different. In fact, I also echo one of the other reviewers' call that you need to provide more of a theoretical grounding, including for those environmental shocks: what does the past literature and existing theory tell us about which way these effects are predicted to go? How would a low-to-high shock look different from a high-to-low shock?

REPLY

Following the advice of all the reviewers, we have provided the theoretical grounding asked for. In our view, the following paragraph indicates the linkages between cognitive abilities and the sustainable management of resources. We also recap our predictions about the importance of cognitive abilities and their salience with respect to changes as follows (starting at line 119):

In the case of environmental change that affects resources, a functional diversity of cognitive abilities should be critical for adapting to negative changes (HL treatment --discussed below), while, perhaps, not as important when environmental change improves a resource (LH treatment--discussed below). The difference in the importance of cognitive abilities may stem from the difference between resource and group dynamics when conditions improve vs. degrade. When conditions improve, there is no need for re-negotiation and one can keep behaving as she did in the past without adverse consequences. On the other hand, when conditions worsen, harvest behavior needs to change in accordance with the new condition of local scarcity. In this context, negotiations about resource appropriation need to happen in

order for the group to continue to manage resources sustainably. For example, in repeated and environmentally stable situations all groups can eventually find optimal solutions (i.e., learning by doing). However, when the ecological system changes, effective mental models of the underlying resource dynamics as well as other individuals' behavior are critical. More effective information processing improves learning and adaptation. Groups composed of individuals with higher g should better adapt to changes in their resource base than groups with lower g, as groups with higher g more readily detect changes in a system and devise rules to match such changes. However, changes in the biophysical resource system often require the re-negotiation of social rules and the communication of new knowledge. Hence, higher ToM should increase a group's ability to work towards a common goal.

* By the way, you need to introduce the low-high and high-low experimental design earlier in the paper: at the moment, I can't tell in which round the resource change happens in either treatment – is it round 3 or 4, or a different round? (It didn't find this information either in "Materials and Methods" or the SI.) Moreover, you mention the treatments before you define them: on page 9, last paragraph, you start a sentence with "In both the high-to-low and low-to-high treatments, ..." without having explained or defined the two treatments earlier.

REPLY

Indeed the reviewer is correct about page 9, however, the treatments are introduced in the first paragraph of the results section. We have amended the text to clarify that participants play the first 3 rounds with a specific growth rate, and the second 3 round with another growth rate. That is, the change happens between rounds 3 and 4. We added the following two statements to the materials and methods section:

See also the following modified text to highlight where changes occurred (starting at line 163) :

In this environment, groups of four harvest tokens for six rounds. Two experimental treatments manipulate the growth function to simulate a perturbation to the resource base of the common pool resource (see Materials and Methods and Supplementary Section 4). In this environment, groups of four harvest tokens for six rounds. Two experimental treatments manipulate the growth function to simulate a perturbation to the resource base of the common pool resource (see Materials and Methods and Supplementary Section 4). In treatment one, groups harvest resources at a high growth rate for three rounds and then experience a sudden decline in the growth rate of the resource and harvest resources with a lower growth rate for another three rounds (HL treatment). In treatment two, the exact opposite sequence occurs (LH treatment). In both treatments, groups harvest tokens with a specific growth rate for three rounds (rounds 1-3), and then with a changed growth rate for another three rounds (rounds 4-6). The change always occurred between rounds 3 and 4.

In order to follow the reviewer's advice, we have also added a clear reference to improving and worsening environmental conditions in the introduction (last paragraph, see changes made in response to the previous comment). We have also reiterated when the change occurs in the supplementary material, section 5, at the end of the first paragraph where we added the following sentence:

"Growth rate was always changed after round three, hence participants played three rounds with either low or high growth rate, and the next three rounds with a changed growth rate."

* Let me clarify something else (or make sure that this is in fact what the authors were intending to get across): in the previous manuscript it sounded like you were contrasting the low-to-high and high-to-low treatments as an empirical exercise, so the interaction in the response letter you provided was the answer to this. But the revision of your main text (and my re-reading of it) suggests that you are not so interested in comparing low-to-high with high-to-low side by side – which is fine, and the revised text doesn't suggest that anymore. HOWEVER, this revision has made clear that you really need to show something about the two different treatments and their (presumably differential) effects on your outcomes. So, instead of doing a between-treatment comparison, you should focus on a "shock to the system" within each treatment and make clear what you expect the effects of low-to-high and high-to-low to be and then test those (see next point).

(2) The second problem is that these low-to-high and high-to-low predictions are not tested explicitly with the regressions. Specifically, you need to be clear what you are saying in the text matches what actually test with regressions. For example, this phrase in the abstract "... even after a shock to the resource system." lets the reader assume that you will test the effect of FIP after (i.e. dummy variable shock=1) a change in environment. This should be tested explicitly in a regression model; at the moment I can't find this anywhere.

REPLY

We are unsure that our and the reviewer's interpretations align. However, we think that our choice of words may have been incorrect, and thus have changed the abstract as follows:

Cognitive abilities underpin the capacity of individuals to build models of their social and ecological environment and make decisions about how to govern resources. Here we test the functional intelligences proposition that functionally diverse cognitive abilities within a group are critical to govern common pool resources. We test this proposition by assessing the effect of cognitive abilities, social and general intelligence, on resource harvesting via two experimental treatments: one in which groups harvest resources before and after a negative shock to the resource base (worsening conditions), and one where groups harvest resources before and after a positive shock to the resource base (improving conditions). Our results indicate that when conditions worsen, groups with high competency in both general and social intelligence are more likely to avoid the depletion of resources and, hence, harvest more resources. Thus, we propose that a functional diversity of cognitive abilities improves how effectively social groups govern common pool resources especially when conditions worsen and groups need to re-assess and re-negotiate their behaviors. We further propose that the relevance of functionally diverse cognitive abilities increases as the complexity of a social-ecological system increases.

* To be clear, from your discussion and specifications of your other models, this does require a three way interaction of the following format: g * ToM * shock for both the low-to-high and high-to-low treatments separately. And three-way interactions are not easy to understand or interpret, so care needs to be taken. In fact, I would recommend you have a table with at least 4 columns (no need for control variables in each of them, maybe add one additional column if you want that includes all control variables at once, but there's no need to have every table have each controls variable added incrementally). So, I ask that you include the following table, first for low-to-high:

Column 1: your standard regressions g * ToM plus a control variable for shock=1 (no interaction with g * ToM yet).

Column 2: same regression ($g * ToM$) but this time only the subsample for which $shock=0$ (i.e. show us the effect of $g*ToM$ before the shock occurred)

Column 3: same regression ($g * ToM$) but this time only the other subsample where $shock=1$ (i.e. after the shock occurred)

Column 4: full three-way interaction model $g * ToM * shock$. This is a big table and will need careful explaining, so don't hand-waive this: instead think carefully what columns 2 and 3 tell you separately and then try to understand and explain column 4 – a three-way interaction is not easy.

(Optional column 5: same as Column 4 but with all the control variables at once if you like.)

* And of course you will need another table like this for high-to-low. Again, I would be surprised if the effects are identical for those two tables since going from low-to-high and high-to-low probably shouldn't be identical, as the experience is completely different. You will need to develop a good theoretical grounding and argument for this (see point 1 above).

REPLY

We do agree with the additional analysis and have performed it and added it both, in the main text and in the supplementary material where we have also added the requested more detailed explanations of the statistical model results. This new information is provided in the Supplementary Section 3.2 and 3.3. In addition to the tables suggested by the reviewer, in this section, we also add a graphical representation of the marginal effects, given that correctly interpreting a 3-way interaction can be challenging, especially when probabilities are nonlinear (i.e. $glm - logit$ link).

We have developed a new analysis of our data following the reviewer's suggestions to observe 3-way interactions. We also explain exactly how we derive our dependent variable ΔT , as well as the Average token collected as % of maximum possible tokens that can be collected (Section 3.1). Further, given that ΔT is a response variable that looks at the overall 6 rounds together, it would have not been possible to assess the three-way interaction on such variable (no variation pre-post ecological change, as $\Delta T = avg$ performance post change – avg performance pre-change. Hence in order to assess the effect of g , ToM and ecological change as well as their interaction we employ avg T , or the average token collected pre-ecological change and post ecological change as a % of the optimal number of tokens that could have been collected (see also Supplementary, section 3.1). In other words, in the supplementary material (Section 3.1) we now explain simulations that we ran to estimate the "optimal" number of tokens that groups could harvest under the different growing conditions. Then, we divided the actual number of tokens collected by groups in each round by the "optimal" for each round. This creates a response variable that measures the percentage of potential tokens harvest by each group per round. The higher the percentage, the better the group was at managing the resource. The lower the percentage, the more the group depleted the resource, missing out on tokens.

This new response variable allows us to assess the three-way interaction of g , ToM and the change in the resource growth rate. The results are reported in full in Supplementary section 3.2, while a summary of figure of the results is also reported in the main text (see Figure 4). The 3-way interaction analysis is consistent with the conclusions drawn in the previous version of the paper.

Finally, to better clarify the marginal effects plots we have also added a "more traditional" representation of marginal effects in the Supplementary Section 3.2, Figure 7.

(3) On another note, I have noticed that you have made some improvements to explain what is going on in your experiment and what the results are. But I think you will need to do more to explain your figures and your findings. The other reviewers and I had both questions about the marginal effects and what g and ToM independently do: it was great that you added Figure 1 to explain this more easily, but please also spend more time to explain what the results mean. At the moment, your paper feels more like "trust that the authors are interpreting the results correctly" than explicitly telling the reader what the coefficients mean and how you arrive at your conclusion – and this is a problem, because as a reader, we can't verify this sort of statement:

"This allows us to track, on average, whether groups push the ecosystem harder (a high positive value), ease their harvest (a negative value), or remain the same (a value of zero) after an ecological change."

Where do you read that from? Please give more evidence and use the regressions to explicitly test your hypothesis before drawing conclusions that are hard to verify (and may or may not be true, I don't know yet).

REPLY

We added more explanations on coefficients and significance in the supplementary material. With respect to the specific quote, a positive value of the coefficients means that groups increase pressure on resources (generally ecosystem) after the perturbation occurs, while a negative sign on the coefficients means that they reduce pressure. With respect to the quote shown, this refers to the actual meaning of the dependent variable ΔT . We have improved the clarity of the quoted passage by adding the rewriting it as follows (starting at line 246).

*To follow up on the results presented above, we ran an OLS regression employing intelligence (g , ToM and $g * ToM$) to predict changes in the percentage of the maximum potential tokens harvested before and after an ecological change: ΔT (see Fig. 3, and Supplementary Table 3). This allows us to track, on average, whether cognitive abilities influence groups' responses to ecological changes by increasing (i.e. harvest more tokens after a change, $\Delta T > 0$), reducing (i.e. harvest fewer tokens after a change, $\Delta T < 0$) or maintaining constant pressure ($\Delta T = 0$). Analogous to above, we ran six models per treatment, one that includes just g , ToM & $g * ToM$ and five that include control variables (see Supplementary Section 3.2 and Table 3 for more details).*

(4) I recommend that you take advantage of the Supplementary Information more: at the moment you have some regression tables there, a bit of methods detail, and the instructions but I think it would pay off to use the space more – you have lots of space in the SI, so please do much of the explaining I asked for above in the SI (with all the details, coefficient interpretations and p-values!) and then take what it is important and transfer it into the main text (with clear cross-references to the corresponding tables in the SI and with clear interpretation of the regressions you are testing there).

REPLY

Indeed, we followed the reviewer's advice and have added added text explaining the main statistical models. In fact, we have re-arranged the statistical model section in the supplementary material separating the main models and their description (Supplementary Section 3.2), the models including the suggested 3-way interactions (Supplementary Section 3.3) and the additional models used to assess the sensitivity of our analysis to different group level aggregation of cognitive abilities (Supplementary Section 3.4). We have not commented in detail on these latter models, as results are consistent except

for two models of Time when using avg ToM. By consistent results, we mean that the signs and thus the directionality of effects are similar. Significance levels however do vary, as well as magnitude of effects (though not drastically).

In sum, this remains an interesting and timely paper but I think the authors need to do more work to (1) clarify their hypotheses, (2) make sure their text matches what they test statistically, (3) use less hand-waiving to articulate their findings and (4) add more detail in the SI to convince readers that they have the evidence they claim they have. I wish the authors good luck for the next revision and I look forward to the next version of the manuscript.

REPLY

We appreciate the reviewer's constructive suggestions, which have strengthened the argument. In response to the reviewer's comments, we have:

- Clarified the hypothesis by highlighting differences between HL and LH conditions;
- Performed new analyses focusing on differences in HL and LH and reported both in the main text (see Figure 4) and in details in the Supplementary Section 3.3;
- Added text to describe the results more thoroughly in the Supplementary (Section 3.2 and 3.3)
- Described the results of key analyses in the main text, with reference to the Supplementary (Section 3)
- Clarified the key variables (notably the ΔT) and added more details on how the dependent variables are calculated in the Supplementary (Section 3.1).

REVIEWERS' COMMENTS:

Reviewer #4 (Remarks to the Author):

Thank you for revising the manuscript. I really enjoyed reading the paper and thought it was much, much better and clearer, especially the work you've done to the abstract, introduction and the additional work in the SI. I felt that the clarification of the worsening/improving conditions was a great way to differentiate between the two conditions and helped to make it easier to understand.

There are no crucial issues outstanding from my perspective – I think this paper will make a very nice contribution to the literature. I have a couple of small suggestions (and they really are suggestions, but I thought I'd bring them up as I noticed them when I read the paper) – but I don't think any required changes are necessary for publication.

1. As mentioned, the abstract is so much clearer than before. My only suggestion would be to have a short sentence about the results for "improving conditions" too: it just feels a bit "incomplete" at the moment since you introduce the notions of worsening and improving conditions but you only directly mention the results for the former. I think it would be helpful to readers if they knew the outcome of improving conditions too.

2. For your literature review on lines 74-87, you have covered Ostrom's seminal work very well; you might also want to include a few more recent papers that have experimentally studied both top-down and bottom-up approaches to public goods governing, e.g. Fehr & Gächter AER (2000); Rand et al. Science (2009); Sutter et al. Rev Econ Stud (2010); Hauser et al. Nature (2014); Güth et al. J Public Economics (2007); Baldassarri & Grossman PNAS (2011); or any of the many other papers on this topic.

3. Page 16 is very helpful and the explanation of your variable coding (rounds 1-3 vs. rounds 4-6) was clear. Thanks for adding this! As a quick suggestion for Figure 4 (but also for the other figures), I would recommend that you add more explanatory figure titles over each panel of the figure. In particular, in Figure 4, your caption with regards to "-b" and "-a" is a bit confusing (although your notion is correct and consistent). My immediate reaction was that "a" referred to panel A when you actually meant the suffix -a for "after". Some readers might be confused by that. Why not label the four panels inside the figure, above each panel, the following way:

"High-to-low Treatment: Before Resource Change" (HLr4-b)

"High-to-low Treatment: After Resource Change" (HLr4-a)

"Low-to-high Treatment: Before Resource Change" (LHr4-b)

"Low-to-high Treatment: After Resource Change" (LHr4-a)

Personally, I would do this for ALL your figures – it will make it easier for people reading over your paper to get a good sense of your results without having to decipher abbreviations.

4. Lines 298-301: "When the ecological change is positive (LH treatment), groups with higher g tend to harvest a greater percentage of potential tokens. This most likely occurs because these groups realize the novel opportunity presented by the decreased likelihood of resource depletion in that environment" – this seems like it's an interesting finding but not very prominent in the current write-up. Like I said above, I think you should include at least a sentence for the "improving conditions" in the abstract, especially because they are a bit different to your main results.

5. I also appreciate that the authors strengthened their SI – it is much easier to read and the tables make more sense in context. Thanks for the extra work!

I look forward to seeing this paper published.

REVIEWERS' COMMENTS:

Reviewer #4 (Remarks to the Author):

Thank you for revising the manuscript. I really enjoyed reading the paper and thought it was much, much better and clearer, especially the work you've done to the abstract, introduction and the additional work in the SI. I felt that the clarification of the worsening/improving conditions was a great way to differentiate between the two conditions and helped to make it easier to understand.

We thank the reviewer for the constructive comments, which clarified the argument and improved the paper.

There are no crucial issues outstanding from my perspective – I think this paper will make a very nice contribution to the literature. I have a couple of small suggestions (and they really are suggestions, but I thought I'd bring them up as I noticed them when I read the paper) – but I don't think any required changes are necessary for publication.

1. As mentioned, the abstract is so much clearer than before. My only suggestion would be to have a short sentence about the results for "improving conditions" too: it just feels a bit "incomplete" at the moment since you introduce the notions of worsening and improving conditions but you only directly mention the results for the former. I think it would be helpful to readers if they knew the outcome of improving conditions too.

We agree with the reviewer and have added a sentence in the abstract describing the results for the improving conditions. The sentence is the following:

"We assess the effect of two cognitive abilities, social and general intelligence, on group performance on a resource harvesting and management game involving either a negative or a positive disturbance to the resource base. Our results indicate that under improving conditions (positive disturbance), groups with higher general intelligence perform better. However, when conditions deteriorate (negative disturbance), groups with high competency in both general and social intelligence are less likely to deplete resources and harvest more."

2. For your literature review on lines 74-87, you have covered Ostrom's seminal work very well; you might also want to include a few more recent papers that have experimentally studied both top-down and bottom-up approaches to public goods governing, e.g. Fehr & Gächter AER (2000); Rand et al. Science (2009); Sutter et al. Rev Econ Stud (2010); Hauser et al. Nature (2014); Güth et al. J Public Economics (2007); Baldassarri & Grossman PNAS (2011); or any of the many other papers on this topic.

Indeed we are familiar with some of the suggested literature. Hence we have added references to the work of Hauser and Baldassari and Grossman when referring to the ability of groups to manage resources sustainably from the bottom up.

3. Page 16 is very helpful and the explanation of your variable coding (rounds 1-3 vs. rounds 4-6) was clear. Thanks for adding this! As a quick suggestion for Figure 4 (but also for the other figures), I would recommend that you add more explanatory figure titles over each panel of the figure. In particular, in Figure 4, your caption with regards to "-b" and "-a" is a bit confusing (although your notion is correct and consistent). My immediate reaction was that "a" referred to panel A when you actually meant the suffix -a for "after". Some readers might be confused by that. Why not label the four panels inside the figure, above each panel, the following way:

"High-to-low Treatment: Before Resource Change" (HLr4-b)

"High-to-low Treatment: After Resource Change" (HLr4-a)

"Low-to-high Treatment: Before Resource Change" (LHr4-b)

"Low-to-high Treatment: After Resource Change" (LHr4-a)

Personally, I would do this for ALL your figures – it will make it easier for people reading over your paper to get a good sense of your results without having to decipher abbreviations.

Indeed we really appreciate this suggestion and have changed the figure titles for Figure 4. However we kept the abbreviation LH and HL in the figure as well given that it is reiterated numerous times within the paper and for the other figures).

4. Lines 298-301: "When the ecological change is positive (LH treatment), groups with higher g tend to harvest a greater percentage of potential tokens. This most likely occurs because these groups realize the novel opportunity presented by the decreased likelihood of resource depletion in that environment" – this seems like it's an interesting finding but not very prominent in the current write-up. Like I said above, I think you should include at least a sentence for the "improving conditions" in the abstract, especially because they are a bit different to your main results.

We added a line in the abstract to highlight results under improving conditions.

5. I also appreciate that the authors strengthened their SI – it is much easier to read and the tables make more sense in context. Thanks for the extra work!

Thanks to you for pushing us to do a much better job. I think the paper has improved (tremendously) thanks to your comments and those of the other two reviewers.

I look forward to seeing this paper published.

We do too! Thanks again for the great insights and comments!